# TVTSv2: Learning Out-of-the-box Spatiotemporal Visual Representations at Scale

## Abstract

The ultimate goal for foundation models is realizing task-agnostic, *i.e.*, supporting out-of-the-box usage without task-specific fine-tuning. Although breakthroughs have been made in natural language processing and image representation learning, it is still challenging for video models to reach it due to the increasing uncertainty of spatiotemporal signals. To ease training, existing works leverage image foundation models' prior knowledge and equip them with efficient temporal modules. Despite the satisfactory fine-tuning performance, we empirically find they fall short of out-of-the-box usage, given the even degraded performance in zero-shot/linear protocols compared to their baseline counterparts. In this work, we analyze the factor that leads to degradation from the perspective of language supervision distortion. We argue that tuning a text encoder end-to-end, as done in previous work, is suboptimal since it may overfit in terms of styles, thereby losing its original generalization ability to capture the semantics of various language registers. The overfitted text encoder, in turn, provides a harmful supervision signal, degrading the video representation. To tackle this issue, we propose a degradation-free pre-training strategy to retain the generalization ability of the text encoder via freezing shallow layers while enabling the task-related semantics capturing in tunable deep layers. As for the training objective, we adopted the transcript sorting task in TVTS (Zeng et al., 2023) incorporated with masking techniques (Li et al., 2023c) to enable scalable training. As a result, we produce a series of models, dubbed TVTSv2, with up to one billion parameters. We achieve new state-of-the-arts on various video benchmarks with a frozen backbone, surpassing the recent ImageBind, InternVideo, *etc*. Code and models will be released publicly.

## 1 Introduction

Learning universal representations that work out of the box[1] on any downstream task is the ultimate goal for foundation models. Inspired by the significant success of large language models (Brown et al., 2020; OpenAI, 2023; Ouyang et al., 2022) in natural language processing, a series of efforts have been devoted to migrating this paradigm to computer vision, *e.g.*, CLIP (Radford et al., 2021), DINO v2 (Oquab et al., 2023), and ImageBind (Girdhar et al., 2023). They pre-train visual models with web-crawled or curated data and enable emerging image-centric applications with pre-trained and frozen visual representations, *e.g.*, zero-shot image classification, and retrieval.

Despite the progress in learning all-purpose image features, there is still a long way to go in the video domain. Given the increasing uncertainty of spatiotemporal signals compared to sole spatial ones, it poses a great challenge for learning task-agnostic and universal video representations. To meet the demand for out-of-the-box usage, a straightforward solution is to scale up the pre-trained video data to cover most distributions. Unfortunately, such an idea is in contrast to the fact that the magnitude of publicly accessible videos is much smaller than images, *e.g.*, HowTo100M (Miech et al., 2019) versus LAION-5B (Schuhmann et al., 2022). Moreover, the nearly quadratic-increased computational overhead in Transformer (Vaswani et al., 2017) is generally unaffordable for training video foundation models at a billion scale.

---

[1]The term "out-of-the-box" indicates that the learned features can be used directly for novel tasks (*e.g.*, zero-shot, linear probing) without tailored fine-tuning.

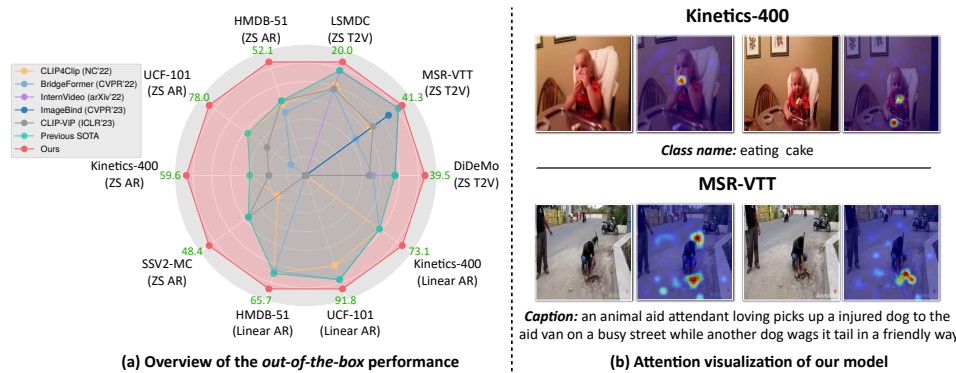

**(a) Overview of the *out-of-the-box* performance**     **(b) Attention visualization of our model**

Figure 1: (a) An overview of the out-of-the-box capability of our learned video representations. ZS, AR, and T2V denote zero-shot, action recognition, and text-to-video retrieval, respectively. (b) Visualization of the self-attention distribution of our pre-trained video model. Taking the corresponding class name or caption as a reference, we observe that our video features can well capture the key spatiotemporal context, which explains our good out-of-the-box ability.

Some works, therefore, turn to exploit the pre-learned spatial prior in image foundation models and adapt them to the video domain by equipping them with temporal reasoning modules, *e.g.*, the recent CLIP-ViP (Xue et al., 2023). Although such methods achieve state-of-the-art performance when fine-tuned on specific downstream tasks, off-the-shelf video representations are not yet suitable for out-of-the-box usage, exhibiting poor zero-shot/linear results and even degradation from CLIP baselines. This may be the reason why frame-level CLIP features are still widely used as video feature extractors (Li et al., 2023a; Dai et al., 2023) despite many efforts that have been made in developing video models. There is a need for real video foundation models that can generate general-purpose video representations.

In this paper, we conduct an in-depth empirical analysis of why previous video models degraded. Intuitively, the performance degradation mainly comes from the overfitting on the relatively small-scale post-pretrain data, sacrificing the generalization ability of the original foundation models. To this end, many existing works focused on efficient visual tuning with adapters (Pan et al., 2022; Yang et al., 2023), while ignoring the potentially overfitted text encoder, which produces distorted text supervision that could, in turn, degrades video representations.

Different from the sole caption knowledge learned from the image foundation models, video data introduces ASR transcripts, which contain temporal dependency and facilitate temporal reasoning, and have been widely adopted in recent literature (Xue et al., 2023; Zeng et al., 2023) to boost spatiotemporal learning. Such supervision provides valuable temporal information but shows a great domain gap from the pre-learned alt-text. Our extensive experiments reveal that *performance degradation in prior methods stems from the text encoder's compromised capabilities of generalizing to various language styles due to the end-to-end fine-tuning with noisy ASR transcripts.* This is a non-trivial issue that few previous papers have yet considered, which not only hinders the proper semantic capture behind the ASR transcripts but also impairs pre-learned language knowledge within the text model, thus negatively affecting video representation learning.

Given the observations, we propose a degradation-free pre-training strategy with a partially frozen text encoder, *i.e.*, the shallow layers are frozen while the deep layers are tunable. With such a strategy, the pre-trained text encoder can generalize well to various language registers with different styles, *e.g.*, ASR transcripts for training, and search queries for downstream retrieval. The new language semantics and structures can be captured in deep layers, thus producing semantically meaningful learning targets to advance out-of-the-box video representation learning. Regarding the training objective, we adopt the pretext task of Turning to Video for Transcript Sorting (TVTS) (Zeng et al., 2023) for its superior performance. To further scale up training to pursue state-of-the-art performance, we adopt the masking strategy without reconstruction (Li et al., 2023c) to enable affordable training with larger backbone architectures. As a result, we successfully trained huge-size models with one billion parameters in total using 80 V100 GPUs in one week.

Finally, we offer a series of pre-trained models, dubbed TVTSv2, from base size to huge size. Compared to the original TVTS (Zeng et al., 2023), we inherit the rich semantic knowledge learned

from CLIP-pretrained models and scale up the models by up to 7 times. The knowledge inheritance is actually not trivial given the degradation of prior methods (Xue et al., 2023; Wang et al., 2021) as we discussed above. As illustrated in Figure 1 (a), our pre-trained model produces all-purpose spatiotemporal visual representations, that can be used for zero-shot/linear video classification, and zero-shot video-text retrieval on various datasets out of the box. Figure 1 (b) effectively highlights how our model precisely captures key spatiotemporal elements, such as the cake being held and the two small dogs near the people, demonstrating the model's remarkable out-of-the-box transferability. It is well noticeable that our TVTSv2 surpasses the recent SOTAs, *i.e.*, ImageBind (Girdhar et al., 2023) and InternVideo (Wang et al., 2022c), on zero-shot video classification and retrieval, despite more data or more modalities they leveraged. Surprisingly, we also achieve comparable performance to DINOv2-g (Caron et al., 2021) on linear K400 with 40% fewer parameters. The encouraging results shed light on the direction of developing general-purpose video foundation models.

## 2 RELATED WORKS

**Out-of-the-box Video Representations.** In out-of-the-box image representation learning, the supervision signal may come from web-crawled images (Oquab et al., 2023), descriptive texts (Radford et al., 2021; Jia et al., 2021), or other modalities (Girdhar et al., 2023), where the second one dominates the literature. Similarly, in the video domain, a bunch of works has made an effort to shift such a paradigm to pursue out-of-the-box video representations. For instance, TVTS (Zeng et al., 2023) adopts a dual-stream architecture and learns fine-grained spatiotemporal representation by resorting to videos for transcript sorting. CLIP4Clip (Luo et al., 2022) stacks a temporal Transformer on top of the origin CLIP to aggregate frame representations. CLIP-ViP (Xue et al., 2023) plugs several video proxy tokens that attend to different frames for temporal summarization. InternVideo (Wang et al., 2022c) inherits UniFormerV2 (Li et al., 2022b) and stacks a cross-modal decoder to enable delicate video-text interaction. Other works either focus on improving vision signals (Xu et al., 2021) or accessing more modalities (Girdhar et al., 2023). However, their improvement in zero-shot and linear probe evaluation is marginal, indicating there is still a long way to reach general-purpose video representation.

**Domain Specialists for Video Tasks.** Besides the video pre-training, another line of research adapted the pre-trained foundation models to the specific video tasks for obtaining domain experts. They reach in-domain gains in two technical routes: (**i**) Designing proper objectives. For instance, Pace (Wang et al., 2020) and SVT (Ranasinghe et al., 2022) learn invariant spatiotemporal characteristics, *e.g.*, motion correspondences of different objects, by aligning clips sampled in a different frame rate. (**ii**) Parameter-efficient tuning. For example, ST-adapter (Pan et al., 2022) and AIM (Yang et al., 2023) plug several tunable spatial and temporal adapters into each attention block, leaving the original parameters frozen. Similarly, visual prompts (Ju et al., 2022; Jia et al., 2022) replace the manually constructed prompt (Radford et al., 2021) with learnable ones to raise instance-specific representations. Nevertheless, such a paradigm is opposite to general-purpose video representation learning, making them inflexible in novel scenarios for real-world applications.

**Scalable Visual Pre-training.** Plenty of work are devoted to improving the model scalability in two aspects: (**i**) Scale-up data. In the video domain, some work harvests from alt-text-video pairs (Bain et al., 2021; Yang et al., 2022) due to the high-quality descriptions, while the available data is limited. The follow-up works utilize ASR transcripts derived from the raw video (Zellers et al., 2021; Miech et al., 2019; Xue et al., 2022) for better scalability. In this work, we reuse the transcript sorting objective (Zeng et al., 2023) for scalable spatiotemporal learning. (**ii**) Reducing computational overheads. The video pre-training is restricted to the quadratically-increased self-attention complexity so far. Masked visual modeling is proposed to improve the efficiency of pre-training, which drops a large portion of visual tokens and reconstructs them given unmasked ones. Although solid results have been reached under a high masking ratio (Feichtenhofer et al., 2022; Tong et al., 2022; Wang et al., 2023), these works are short of out-of-the-box transferability due to train-test mismatch. Recently, FLIP (Li et al., 2023c) directly conducts image-text contrastive pre-training based on masked visual tokens and achieves favorable results. Inspired by this work, we train the models with up to one billion parameters by properly incorporating masking.

## 3 METHOD

In this section, we introduce how to learn out-of-the-box video representations at scale without the performance degradation of the image pre-trained knowledge. Our main framework is illustrated in

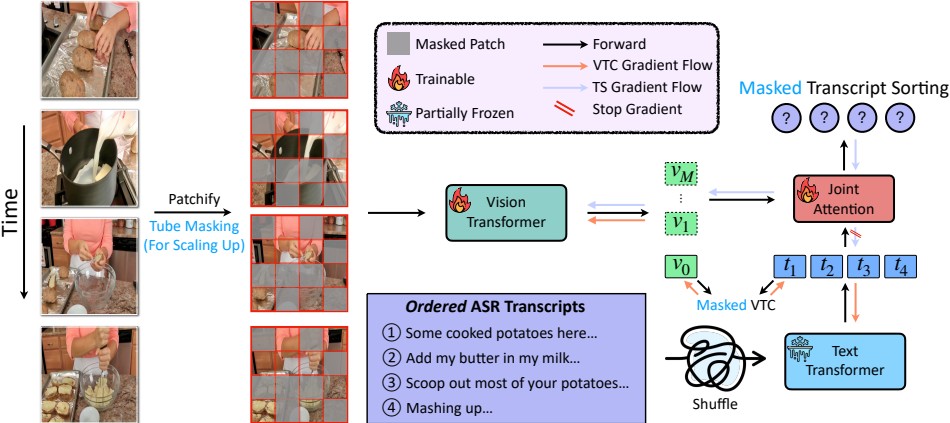

Figure 2: Our training framework. A large portion of patches from sampled frames is first dropped out via random tube masking before being sent into a Vision Transformer equipped with divided space-time attention for encoding video representations. The corresponding ASR transcripts are shuffled and embedded by a partially frozen text encoder. The disposable joint attention (only for training) is performed among all video and text representations for predicting the transcripts' chronological order, formulated by a $K$-way classification objective.

Figure 2, where the trained Vision Transformer is deployed for extracting video features that work out of the box on downstream tasks. We will introduce our partially frozen training strategy to avoid knowledge degradation in Section 3.1, our model architectures in Section 3.2, our masking mechanism to enable scalable pre-training in Section 3.3, and our training objectives in Section 3.4.

## 3.1 DEGRADATION-FREE PRE-TRAINING WITH PARTIALLY FROZEN TEXT ENCODER

Learning video representations with the assistance of text dominates recent literature (Bain et al., 2021; Ge et al., 2022a; Xue et al., 2023). There are two main lines of research: **(i)** Learning video features under alt-text supervision, *i.e.*, video caption. The alt-text is generally clean but hard to scale up, *e.g.*, WebVid-10M (Bain et al., 2021). **(ii)** Using the ASR transcripts as supervision which are naturally tied with the video and easy to scale up, *e.g.*, YT-Temporal (Zellers et al., 2021). Despite some noise, transcripts generally provide temporal dependency that facilitates temporal reasoning.

Existing methods relying solely on alt-text (Bain et al., 2021; Ge et al., 2022a;b) achieve marginal improvements in video tasks due to limited data availability. Subsequent works (Zeng et al., 2023; Xue et al., 2023) focus on effectively utilizing large-scale ASR transcripts to enhance temporal modeling in downstream tasks. Although fine-tuning yields impressive results, their out-of-the-box video representations are degraded compared to pre-trained image counterparts, meaning the zero-shot capability is inferior to simply aggregating frame-level image features. Recently, CLIP-ViP (Xue et al., 2023) attributed this to the domain gap between ASR transcripts and downstream captions and proposed using OFA-generated captions (Wang et al., 2022b) for training. However, these generated captions do not effectively improve out-of-the-box video representation learning. Our detailed empirical analysis is presented below.

**An empirical study of video degradation.** To identify the cause of video degradation and develop a reliable solution, we meticulously design experiments with six different pre-training settings. The out-of-the-box capability of the learned representations is evaluated using two zero-shot metrics: top-1 action recognition accuracy on Kinetics-400 and text-to-video retrieval recall@1 on DiDeMo.

As shown in Table 1 L1-L2, the CLIP-ViP trained with OFA-generated captions experiences performance degradation compared to the CLIP baseline. To analyze the domain gap impact, we create two baselines from CLIP-ViT-B/32, *i.e.*, $M_{ASR-Full}$ (using YT-Temporal) and $M_{Alt-Full}$ (using WebVid-2.5M), both optimized solely by the Video-Text Contrastive objective. Table 1 L3-L4 reveals that $M_{ASR-Full}$ degrades more than CLIP-ViP on DiDeMo, while $M_{Alt-Full}$ achieves significant gains, particularly on Kinetics-400. When jointly trained on ASR and alt-text corpus as $M_{All-Full}$ (Table 1 L5), it slightly outperforms CLIP-ViP due to better text quality but remains inferior to the CLIP baseline on DiDeMo, suggesting that *merely reducing the domain gap offers limited improvements and does not fully address the degradation issue.*

Table 1: The top-1 accuracy on Kinetics-400 and recall@1 on DiDeMo *w.r.t.* different settings. ZS, AR, and T2V denote zero-shot, action recognition, and text-to-video retrieval respectively. HD-VILA-OFA denotes the OFA-generated caption for videos in HD-VILA-100M. The tests are done with the ViT-B/32 models, and all variants are trained by the Video-Text Contrastive objective only.

| No. | Method | ASR | Alt-text | Text Encoder Tuning Strategy | Kinetics-400 (ZS AR) | DiDeMo (ZS T2V) |
|-----|--------|-----|----------|------------------------------|----------------------|-----------------|
| L1 | CLIP | ✗ | ✗ | ✗ | 42.7 (+0.0) | 24.7 (+0.0) |
| L2 | $M_{ASR-Full}$ | HD-VILA | HD-VILA-OFA | ☀ Full | 37.0 (-5.7) | 22.6 (-2.1) |
| L3 | $M_{ASR-Full}$ | YT-Temporal | ✗ | ☀ Full | 45.7 (+3.0) | 19.3 (-5.4) |
| L4 | $M_{Alt-Full}$ | ✗ | WebVid-2.5M | ☀ Full | 48.6 (+5.9) | 28.4 (+3.7) |
| L5 | $M_{All-Full}$ | YT-Temporal | WebVid-2.5M | ☀ Full | 47.9 (+5.2) | 24.2 (-0.5) |
| L6 | $M_{All-Frozen}$ | YT-Temporal | WebVid-2.5M | ❄ Frozen | 45.2 (+2.5) | 21.9 (-2.8) |
| L7 | $M_{ASR-Partial}$ | YT-Temporal | ✗ | ⛅ Partial | 48.1 (+5.4) | 26.8 (+2.1) |
| L8 | $M_{All-Partial}$ | YT-Temporal | WebVid-2.5M | ⛅ Partial | **50.8** (+8.1) | **29.8** (+5.1) |

Considering that $M_{Alt-Full}$ achieves the best performance while $M_{All-Full}$ falls short, we draw two conclusions: (**i**) Knowledge from alt-text-image pairs is more suitable for zero-shot video domain inference, and (**ii**) The end-to-end tuning approach for the ASR corpus is suboptimal, causing catastrophic forgetting in the text branch and negatively impacting out-of-the-box performance. To test this, we retrain $M_{All-Full}$ with a fully frozen text encoder as $M_{All-Frozen}$. However, the inferior results in Table 1 L6 suggest that the pre-trained and frozen text encoder fails to fails to correctly capture the language semantics and produces harmful learning targets.

Given the above observations, we come up with a simple yet effective training strategy, namely partially frozen (PF), which freezes the shallow layers of the text encoder. Such a strategy promotes the text encoder's generalizability *w.r.t.* different language registers with varying styles. The semantics and structures of ASR transcripts are captured in deep layers, producing semantically meaningful supervision signals to facilitate out-of-the-box video representation learning. In our practice, we freeze the first three-quarters of layers, leaving other layers trainable. As demonstrated in Table 1 L7-L8, the partially frozen models, *i.e.*, $M_{ASR-Partial}$ and $M_{All-Partial}$, significantly outperform their fully tuned counterparts (L3 and L5). Furthermore, $M_{All-Partial}$ achieves the best performance, suggesting that *pre-training with a partially frozen text encoder could well preserve the knowledge of the CLIP-pretrained model and unleash it for learning strong out-of-the-box video representations.*

## 3.2 MODEL ARCHITECTURES

To inherit the knowledge of pre-trained image foundation models, *i.e.*, CLIP, we apply the dual-stream framework, which consists of a visual encoder and a text encoder. We extend the visual encoder to the video domain by equipping it with divided space-time attention following the practice in Bertasius et al. (2021).

**Video Encoder.** We adopt the standard Vision Transformer (ViT) (Dosovitskiy et al., 2021) with $L_V$ layers as the video encoder and add divided space-time attention (Bertasius et al., 2021) in each layer for spatiotemporal reasoning. Given $T$ frames sampled from a video with a resolution of $3 \times H \times W$, we split each frame into patches of size $P \times P$, then project them to a token sequence $v \in \mathbb{R}^{TN \times D_V}$, where $N = HW/P^2$ and $v_i \in \mathbb{R}^{D_V}$ denotes the $i$-th token with a dimension $D_V$. Next, learnable spatial and temporal positional embeddings, denoted as $\mathbf{E}^s \in \mathbb{R}^{N \times D_V}$ and $\mathbf{E}^t \in \mathbb{R}^{T \times D_V}$, are added to each token, *i.e.*,

$$v_{x,y}^{(0)} = v_{x,y} + \mathbf{E}_x^s + \mathbf{E}_y^t \tag{1}$$

where $v_{x,y}$ denotes the *x*-th token sampled from the *y*-th frame. All *x*-th tokens sampled from different frames are given the same spatial positional embedding, and all tokens belonging to the *y*-th frame are given the same temporal positional embedding. Finally, a learnable [CLS] token $v_0 \in \mathbb{R}^{D_V}$ is concatenated at the beginning of the sequence, and we perform divided space-time attention across the $L_V$ layers. The [CLS] token output by the $L_V$-th layer is further projected to a $D$-dimensional shared space and used as our final video representation for out-of-the-box usage.

**Text Encoder.** The text encoder contains $L_T$ stacked Transformer blocks. Following CLIP (Radford et al., 2021), we adopt the casual attention mechanism. Similar to the visual branch, we project the [CLS] token $t_0$ output by the last layer to the $D$-dimensional shared space as the final representation.

## 3.3 SCALABLE PRE-TRAINING WITH MASKED VIDEO ENCODER

We aim to pursue scalable video representation learning with larger backbone architectures in order to achieve better out-of-the-box capabilities. However, the token length, *i.e.*, $T * N$, is always a bottleneck for scaling up video training due to spatiotemporal attention's drastically increased computational budget. The recent VideoMAEs (Tong et al., 2022; Wang et al., 2023) improve training efficiency by masking a high proportion of patches, significantly reducing the token length. A similar practice was observed in image pre-training, *i.e.*, FLIP (Li et al., 2023c). Inspired by their success, we experiment with video pre-training at high mask ratios without reconstruction.

Specifically, we employed the tube masking approach introduced by VideoMAE (Tong et al., 2022) because it's compatible with divided space-time attention and can effectively diminish temporal redundancy. Essentially, this method randomly blocks $\rho\%$ of patches from the same position across various frames, resulting in a token length of $TN(1-\rho)$. For this study, we set $\rho \geq 50\%$, translating to at least half of the computational budget being reduced. This ensures our approach can be scaled to models with billions of parameters.

## 3.4 TRAINING OBJECTIVES

**Transcript Sorting (TS).** Sorting the unordered transcripts given ordered video patches has been proven effective for learning sophisticated spatiotemporal interaction (Zeng et al., 2023). In this work, we further reveal the scalability of TS for training large models under a high mask ratio. Given translated words and their timestamps $\{w_i, a_i\}_{i=1}^{N_w}$, where $w_i$ denotes the *i*-th word, $a_i$ denotes its corresponding timestamp (in seconds), and $N_w$ denotes the word number, we sample $K$ transcript segments $\{T_k\}_{k=1}^{K}$ of length $l$ (in seconds) with an interval of 1s between consecutive segments:

$$L_k = L_{\text{start}} + (k - 1) * (l + 1)$$
$$T_k = \{w_i | a_i \in [L_k, L_k + l]\} \tag{2}$$

where $L_{\text{start}}$ denotes a randomly picked starting time, and $L_k$ denotes the beginning time of the *k*-th segments. Then we randomly shuffle the $K$ segments to $\{T_{o_i}\}_{i=1}^{K}$, where $T_{o_i}$ denotes the *i*-th segment in the shuffled sequence corresponding to the ground-truth chronological order $o_i$. As for frame sampling, we follow TSN (Wang et al., 2016) to divide the overall interval, *i.e.*, $[L_1, L_K + l]$, into $T$ equal-space intervals and randomly sample 1 frame per interval.

After encoding sampled transcript segments and frames, we concatenate the text [CLS] tokens $\{t_0^i\}_{i=1}^{K}$ with all unmasked video tokens $\{v_i\}_{i=0}^{TN\rho}$ and perform joint attention across them. Next, we send the attend text [CLS] tokens into a $K$-way classifier and predict their orders separately. Finally, the transcript sorting objective is formulated as a cross-entropy:

$$\mathcal{L}_{\text{TS}} = \frac{1}{K} \sum_{i=1}^{K} - \log \frac{\exp(p_{o_i}^i)}{\sum_{j=1}^{K} \exp(p_j^i)} \tag{3}$$

where $p^i \in \mathbb{R}^K$ denotes the prediction for the *i*-th transcript segment in the shuffled sequence, $p_j^i$ indicates the probability that its chronological order is $j$, and $o_i$ represents the ground-truth order.

It is worth noting that the model may learn shortcuts, *e.g.*, natural language order, without attending to visual information. We prevent such cases from hurting training by stopping gradients flowing to the text encoder, which forces the video model to provide well-learned spatiotemporal context.

**Video-Text Contrastive.** We adopt the widely-used Video-Text Contrastive (VTC) as the basic objective for semantic alignment, which is formulated as:

$$\mathcal{L}_{\text{VTC}} = \text{NCE}(\bar{t}, v_0) + \text{NCE}(v_0, \bar{t})$$
$$s.t. \quad \text{NCE}(q, k) = - \log \frac{\exp(q^T k_+ / \tau)}{\sum_{i=1}^{B} \exp(q^T k_i / \tau)} \tag{4}$$

where $\tau$ denotes the temperature, $\bar{t}$ denotes the averaged text [CLS] token, *i.e.*, $\bar{t} = \frac{1}{K} \sum_{i=1}^{K} t_0^i$, and $v_0$ denotes the video [CLS] token. Note that $v_0$ only attends to the unmasked patches during encoding. Our overall training objective is $\mathcal{L} = \mathcal{L}_{\text{VTC}} + \lambda \mathcal{L}_{\text{TS}}$, where $\lambda$ is a hyperparameter.

Table 2: The **zero-shot text-to-video retrieval** results. † denotes using post-processing DSL (Cheng et al., 2021). ∗ means accessing extra modalities, *e.g.*, audio. The underlined number indicates absolute SOTA. Single-stream models are de-emphasized.

| Method | Venue | MSR-VTT | | | | DiDeMo | | | | LSMDC | | | |
|---|---|---|---|---|---|---|---|---|---|---|---|---|---|
| | | R@1 | R@5 | R@10 | MdR | R@1 | R@5 | R@10 | MdR | R@1 | R@5 | R@10 | MdR |
| *Non-CLIP models* | | | | | | | | | | | | | |
| VideoCLIP (Xu et al., 2021) | EMNLP'21 | 10.4 | 22.2 | 30.0 | - | 16.6 | 46.9 | - | - | - | - | - | - |
| Frozen (Bain et al., 2021) | ICCV'21 | 18.7 | 39.5 | 51.6 | 10.0 | 21.1 | 46.0 | 56.2 | 7.0 | 9.3 | 22.0 | 30.1 | 51.0 |
| ALPRO (Li et al., 2022a) | CVPR'22 | 24.1 | 44.7 | 55.4 | - | 23.8 | 47.3 | 57.9 | - | - | - | - | - |
| VIOLET (Fu et al., 2021) | arXiv'22 | 25.9 | 49.5 | 59.7 | - | 23.5 | 49.8 | 59.8 | - | - | - | - | - |
| BridgeFormer (Ge et al., 2022a) | CVPR'22 | 26.0 | 46.4 | 56.4 | 7.0 | 25.6 | 50.6 | 61.1 | 5.0 | 12.2 | 25.9 | 32.2 | 42.0 |
| OmniVL (Wang et al., 2022a) | NeurIPS'22 | 34.6 | 58.4 | 66.6 | - | 33.3 | 58.7 | 68.5 | - | - | - | - | - |
| *CLIP-B/32* | | | | | | | | | | | | | |
| CLIP (Radford et al., 2021) | ICML'21 | 30.6 | 54.4 | 64.3 | 4.0 | 24.7 | 49.3 | 60.9 | 6.0 | 13.6 | 27.9 | 35.5 | 32.0 |
| CLIP-straight (Portillo et al., 2021) | MCPR'21 | 31.2 | 53.7 | 64.2 | 4.0 | - | - | - | - | 11.3 | 22.7 | 29.2 | 56.5 |
| CLIP4Clip (Luo et al., 2022) | NC'22 | 32.0 | 57.0 | 66.9 | 4.0 | - | - | - | - | 15.1 | 28.5 | 36.4 | 28.0 |
| BridgeFormer (Ge et al., 2022a) | CVPR'22 | 33.2 | 58.0 | 68.6 | 4.0 | - | - | - | - | 15.5 | 30.7 | 38.7 | 22.0 |
| CLIP-ViP (Xue et al., 2023) | ICLR'23 | 29.0 | 51.2 | 61.3 | 5.0 | 22.6 | 43.9 | 56.4 | 7.0 | 11.3 | 25.3 | 31.3 | 38.0 |
| Ours-B/32 | - | 34.5 | **58.5** | 67.7 | 3.5 | 31.2 | 56.9 | 68.3 | 4.0 | **16.1** | 30.6 | **38.7** | 25.0 |
| Ours-B/32† | - | **36.4** | 58.0 | **69.0** | **3.0** | **37.0** | **61.6** | **70.9** | **3.0** | 15.6 | **31.8** | 38.6 | **22.0** |
| *CLIP-B/16* | | | | | | | | | | | | | |
| CLIP (Radford et al., 2021) | ICML'21 | 31.8 | 53.9 | 64.5 | 4.0 | 27.7 | 51.0 | 62.5 | 5.0 | 15.2 | 29.7 | 37.6 | 25.0 |
| CLIP-ViP (Xue et al., 2023) | ICLR'23 | 31.7 | 53.8 | 63.2 | 4.0 | 24.6 | 50.7 | 59.7 | 5.0 | 12.5 | 26.1 | 33.3 | 39.0 |
| UMT-B (Li et al., 2023b) | ICCV'23 | 29.6 | 52.8 | 61.9 | - | 33.4 | 58.3 | 67.0 | - | 16.8 | 30.5 | 37.6 | - |
| Ours-B/16 | - | 35.9 | 61.2 | 71.3 | 3.0 | 33.4 | 60.1 | 70.6 | 3.0 | 16.9 | 31.5 | 38.2 | 22.0 |
| Ours-B/16† | - | **37.8** | **62.9** | **72.4** | **3.0** | **39.0** | **63.9** | **72.6** | **3.0** | **18.3** | **33.7** | **41.9** | **19.0** |
| *Larger Models* | | | | | | | | | | | | | |
| ImageBind∗ (Girdhar et al., 2023) | CVPR'23 | 36.8 | 61.8 | 70.0 | - | - | - | - | - | - | - | - | - |
| InternVideo (Wang et al., 2022c) | arXiv'22 | 40.0 | 65.3 | 74.1 | 2.0 | 31.5 | 57.6 | 68.2 | 3.0 | 17.6 | 32.4 | 40.2 | 23.0 |
| UMT-L (Li et al., 2023b) | ICCV'23 | 33.3 | 58.1 | 66.7 | - | 34.0 | 60.4 | 68.7 | - | 20.0 | 37.2 | 43.7 | - |
| Ours-L/14 | - | 36.9 | 62.0 | 72.9 | 3.0 | 33.9 | 59.9 | 71.0 | 3.0 | 17.1 | 32.3 | 40.6 | 20.0 |
| Ours-L/14† | - | 40.1 | 62.6 | 73.5 | 3.0 | 38.3 | 62.6 | 72.5 | 3.0 | 19.3 | 36.2 | 43.1 | 17.0 |
| Ours-H/14 | - | 38.2 | 62.4 | 73.2 | 3.0 | 34.6 | 61.9 | 71.5 | 3.0 | 17.3 | 32.5 | 41.4 | 20.0 |
| Ours-H/14† | - | **41.3** | **63.0** | **74.0** | **2.0** | **39.5** | **63.6** | **73.1** | **3.0** | **20.0** | **37.8** | **48.6** | **11.0** |

# 4 EXPERIMENTS

## 4.1 EXPERIMENTAL SETUP

**Pre-training Datasets.** We jointly pre-train our model on two datasets: (a) **YT-Temporal** (Zellers et al., 2021) contains 6M YouTube videos with ASR transcribed words and timestamps. (b) **WebVid-2.5M** (Bain et al., 2021) contains 2.5M alt-text-video pairs. Since the timestamps are unavailable, we only perform VTC on it.

**Text-to-Video Retrieval.** We evaluate *zero-shot* performance of our model on three benchmarks: (a) **MSR-VTT** (Xu et al., 2016a). (b) **DiDeMo** (Anne Hendricks et al., 2017). (c) **LSMDC** (Rohrbach et al., 2015). The Recall@K (R@K) and Median Rank (MdR) are reported as the evaluation metric.

**Action Recognition and Anomaly Detection.** Six benchmarks are used for evaluating *zero-shot* performance: (a) **HMDB-51** (Kuehne et al., 2011), (b) **UCF-101** (Soomro et al., 2012), (c) **Kinetics-400** (Kay et al., 2017) (K400), (d) **Kinetics-600** (Carreira et al., 2018) (K600), (e) **SSV2** (Goyal et al., 2017), (f) **UCF-Crime** (Sultani et al., 2018) (Crime). Following Xue et al. (2023), we use the prompt template "a person [CLASS]" for the first four datasets and evaluate SSV2 on a multi-choice setting, namely **SSV2-MC**. As for the anomaly detection benchmark, *i.e.*, UCF-Crime, we use the prompt template "a video of [EVENT]" to recognize the anomaly events. See Appendix for details.

## 4.2 IMPLEMENTATION DETAILS

We inherit weights from CLIP (Radford et al., 2021) for the standard ViT modules and initialize the parameters of time attention with zeros (Bertasius et al., 2021). We sample $K = 4$ transcript segments for the transcript sorting task, and $T = 12$ frames with an input resolution of $224 \times 224$ for pre-training and downstream tasks. The masking ratio $\rho$ is set to $50\%$ for ViT-B/16 and $70\%$ for ViT-H/14, and we do not mask patches for ViT-B/32. All hyper-parameters are listed in Appendix.

## 4.3 MAIN RESULTS

**Text-to-Video Retrieval.** The zero-shot retrieval results are reported in Table 2. Compared to previous dual-stream SOTA, our model reaches an absolute gain of $1.4\%$, $6.9\%$ and $1.8\%$ in terms of R@1 on MSR-VTT, DiDeMo, and LSMDC, respectively. Using post-processing techniques like

Table 3: The **zero-shot action recognition and anomaly detection** results of top-1 accuracy. The underlined number indicates absolute SOTA. Single-stream models are de-emphasized. "Our impl." denotes we use official pre-trained weights for evaluation. $\text{UMT}_{\text{dual}}$ refers to dual-stream style inference using UMT's vision and text encoder.

| Method | Venue | Cite From | Params | HMDB-51 | UCF-101 | K400 | K600 | SSV2-MC | Crime |
|---|---|---|---|---|---|---|---|---|---|
| *Non-CLIP models* | | | | | | | | | |
| MTE (Xu et al., 2016b) | ECCV'16 | X-CLIP | $<200$M | 19.7 | 15.8 | - | - | - | - |
| ASR (Wang et al., 2017) | ECML'17 | X-CLIP | | 21.8 | 24.4 | - | - | - | - |
| ZSECOC (Qin et al., 2017) | CVPR'17 | X-CLIP | | 22.6 | 15.1 | - | - | - | - |
| UR (Zhu et al., 2018) | CVPR'18 | X-CLIP | | 24.4 | 17.5 | - | - | - | - |
| TS-GCN (Gao et al., 2019) | AAAI'19 | X-CLIP | | 23.2 | 34.2 | - | - | - | - |
| E2E (Brattoli et al., 2020) | CVPR'20 | X-CLIP | | 32.7 | 48.0 | - | - | - | - |
| ER-ZSAR (Chen et al., 2021) | ICCV'21 | X-CLIP | | 35.3 | 51.8 | - | 42.1 | - | - |
| ClipBert (Lei et al., 2021) | CVPR'21 | BridgeFormer | | 20.0 | 27.5 | - | - | - | - |
| Frozen (Bain et al., 2021) | ICCV'21 | BridgeFormer | | 27.5 | 45.4 | - | - | - | - |
| BridgeFormer (Ge et al., 2022a) | CVPR'22 | BridgeFormer | | 38.0 | 51.1 | - | - | - | |
| *CLIP-B/16* | | | | | | | | | |
| CLIP (Radford et al., 2021) | ICML'21 | Our impl. | $<200$M | 43.2 | 68.9 | 48.0 | 62.4 | 29.6 | 25.5 |
| ActionCLIP (Wang et al., 2021) | arXiv'21 | X-CLIP | | 40.8 | 58.3 | - | - | - | - |
| X-CLIP (Ni et al., 2022) | ECCV'22 | X-CLIP | | 44.6 | 72.0 | - | 65.2 | - | - |
| CLIP-ViP (Xue et al., 2023) | ICLR'23 | Our impl. | | 41.2 | 58.9 | 37.6 | 46.7 | 35.5 | 19.1 |
| $\text{UMT}_{\text{dual}}$ (Li et al., 2023b) | ICCV'23 | Our impl. | | 32.7 | 46.4 | 34.1 | 44.0 | 23.5 | - |
| Ours-B/16 | - | - | | **50.4** | **69.8** | **54.3** | **68.1** | **42.1** | **28.2** |
| *Larger Models* | | | | | | | | | |
| ImageBind (Girdhar et al., 2023) | CVPR'23 | ImageBind | 632M (V)
354M (T) | - | - | 50.0 | - | - | - |
| X-Florence (Ni et al., 2022) | ECCV'22 | X-CLIP | 637M (V)
256M (T) | 48.4 | 73.2 | - | 68.8 | - | - |
| Ours-H/14 | - | - | 632M (V)
354M (T) | **52.1** | **78.0** | **59.6** | **73.2** | **48.4** | **30.9** |

Table 4: The zero-shot text-to-video retrieval results *w.r.t.* different objectives. "sg" denotes stopping gradients.

| Name | $\mathcal{L}_{\text{VTC}}$ | $\mathcal{L}_{\text{TS}}$ | sg | MSR-VTT | | DiDeMo | | LSMDC | |
|---|---|---|---|---|---|---|---|---|---|
| | | | | R@1 | R@5 | R@1 | R@5 | R@1 | R@5 |
| CLIP | n/a | n/a | n/a | 30.6 | 54.4 | 24.7 | 49.3 | 13.6 | 27.9 |
| $\text{M}_{\text{base}}$ | ✓ | × | n/a | 33.0 | 56.7 | 29.8 | 55.5 | 15.2 | 29.3 |
| $\text{M}_{\text{w/o sg}}$ | ✓ | ✓ | × | 31.9 | 56.2 | 29.2 | 54.9 | 14.9 | 28.9 |
| $\text{M}_{\text{ours}}$ | ✓ | ✓ | ✓ | **34.5** | **58.5** | **31.2** | **56.9** | **16.1** | **30.6** |

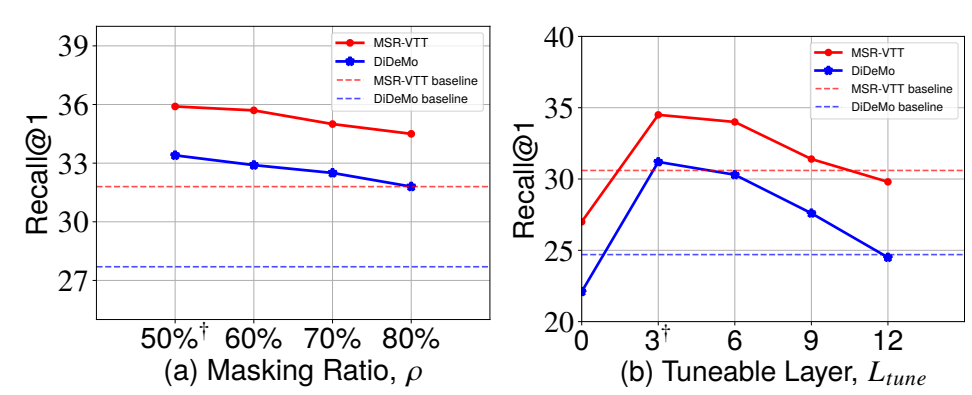

Table 5: The sensitivity of (a) masking ratio $\rho$, (b) tuneable layer $L_{\text{tune}}$. Default settings are marked with †.

DSL (Cheng et al., 2021) further boosts performance. It is worth noting that we achieve comparable performance with single-stream models, *e.g.*, InternVideo (Wang et al., 2022c), which uses a fusion module to encode multimodal entangled representations, thereby incapable of out-of-the-box usage, *i.e.*, producing video-only representations. Moreover, the solid result further verifies the effectiveness of the degradation-free pre-training strategy, especially given the fact that CLIP-ViP (Xue et al., 2023) degrades and CLIP4Clip (Luo et al., 2022) only makes marginal improvement.

**Action Recognition and Anomaly Detection.** We report the zero-shot action recognition results in Table 3. Our model brings significant improvements, *i.e.*, 8.9%, 9.1%, 11.6%, 10.8% and 12.9% absolute gain on HMDB-51, UCF-101, K400, K600, and SSV2-MC, respectively. The prompt-based method, *i.e.*, ActionCLIP (Wang et al., 2021), also degrades on this test, possibly due to overfitting to manually constructed templates. When it turns to larger models, we even surpass X-Florecnce (Ni et al., 2022), a single-stream model with comparable video encoder parameters, by a large margin, revealing the superior scalability of our training paradigm. To make a fair and meaningful comparison with single-stream models, we remove the cross-modal encoder of UMT and directly use the features outputted by the text and visual branch, namely $\text{UMT}_{\text{dual}}$. The worst performance indicates that it is incapable of out-of-the-box usage. Even for the challenging SSV2-MC that contains various motion dynamics, our model still achieves promising results, solidifying the contribution of masked transcript sorting that promotes fine-grained spatiotemporal representation learning. As for anomaly detection, our models also made remarkable improvements compared to the CLIP baseline and CLIP-ViP on Crime, unveiling their adaptability and generalizability. We also provide the liner classification results in Appendix, where we beat either self-supervised or language-guided models.

### 4.4 Ablation Study

Considering efficiency, all ablations are based on the ViT-B/32 model if no specified.

**Training Objectives.** We first analyze the impact of each training objective in Table 4 and reach the following conclusions: (**i**) $M_{base}$ outperforms the CLIP baseline, implying that both ASR transcripts and alt-texts are reliable supervision that benefits video representation learning. (**ii**) $M_{ours}$ outperforms $M_{base}$, while $M_{w/o\ sg}$ underperforms. It indicates that without stopping gradients flowing to the text encoder, the model turns to learn shortcuts, in other words, optimize the transcript representation to ease the sorting task. The wrong optimization direction harms training efficiency. Instead, $M_{ours}$ encourages enhancing spatiotemporal representation to provide enough knowledge for transcript sorting, remarkably improving the performance.

**Masking Ratio.** We report the Recall@1 under different masking ratio $\rho$ in Table 5 (a) based on the ViT-B/16 model. Given the same training steps, smaller $\rho$s lead to better performance due to the more learned tokens. We imply that over-masking makes it too hard for the model to capture enough visual semantics despite more training efficiency. As a result, for different scales of models, we need to find a proper masking ratio that could trade off the performance and training cost.

**Tuneable Layer.** The performance in terms of different tuneable layers $L_{tune}$ (count from the last layer) for the text encoder is reported in Table 5 (b). $L_{tune} = 0$ equals to fully freeze the text encoder, and $L_{tune} = 12$ means fully fine-tuning. Both $L_{tune} = 0$ and $L_{tune} = 12$ lag far behind partially freezing the text encoder, verifying the effectiveness of the proposed degradation-free pre-training strategy, which preserves the image foundation model's pre-learned knowledge while unleashing it to learn powerful out-of-the-box spatiotemporal representation.

**Generaliability.** We further transfer our training framework to HowTo100M Miech et al. (2019), which consists of 1.2M videos paired with ASR transcripts, and employ CLIP-B/16 with a masking ratio of 50% for training. Table 6 reports the results, with FT representing full tuning and PF denoting partially freezing. The effectiveness of partial freezing on HowTo100M is evident when comparing $M_{HT-PF}$ and $M_{HT-FT}$. For FT variants, $M_{YT-FT}$ underperforms $M_{HT-FT}$ due to greater ASR noise and more severe text distortion. However, when using partial freezing, $M_{YT-PF}$ significantly outperforms $M_{HT-PF}$ as it benefits from a larger dataset. This suggests that our partial freezing strategy is advantageous for scaling up video training data and practical for pre-training.

Table 6: The zero-shot R@1 on DiDeMo and top-1 accuracy on Kinetics-400 *w.r.t.* different datasets and training strategies. WV, YT, and HT are short for WebVid, YT-Temporal, and HowTo100M, respectively. FT denotes full tune, and PF denotes partial freezing.

| Name | Alt-texts | ASR Transcripts | ASR Domain | DiDeMo | Kinetics-400 |
|---|---|---|---|---|---|
| $M_{YT-FT}$ | WV2.5M | YT-5M | Open | 27.4 | 49.7 |
| $M_{YT-PF}$ | | | | 33.4 | 54.3 |
| $M_{HT-FT}$ | WV2.5M | HT-1.2M | Instructional | 30.8 | 51.7 |
| $M_{HT-PF}$ | | | | 32.2 | 52.9 |

Similar patterns are observed across different vision models. For instance, we replace ViT with ResNet-50, keep the text branch unchanged, and train the models for two epochs. Table 7 presents the results, showing that ResNet50-FT trails ResNet50-PF, which employs the partially frozen strategy, by a considerable margin. The differences are expected to be even more significant when training for longer epochs. These results demonstrate the strong generalizability of our proposed training scheme.

Table 7: The zero-shot evaluation results using ResNet-50 as the vision model. FT denotes full tune, and PF denotes partial freezing.

| Method | MSR-VTT | | | Kinetics-400 | | SSV2-MC | |
|---|---|---|---|---|---|---|---|
| | R@1 | R@5 | MdR | Top-1 | Top-5 | Top-1 | Top-5 |
| ResNet50$_{FT}$ | 17.0 | 37.6 | 12.0 | 27.7 | 56.2 | 22.6 | 46.8 |
| ResNet50$_{PF}$ | 20.0 | 41.6 | 10.0 | 31.6 | 58.9 | 24.9 | 49.9 |

## 5 CONCLUSION

In this work, we pursue out-of-the-box spatiotemporal visual representations with a newly proposed TVTSv2. Compared to TVTS (Zeng et al., 2023), we introduce a degradation-free pre-training strategy to solve the representation degradation from the pre-trained image foundation model, which is actually non-trivial given the failure of previous models (Xue et al., 2023; Wang et al., 2021). We further adopt the masking technique (Li et al., 2023c) to improve scalability. With TVTSv2, we train several models with up to one billion parameters, achieving SOTA results in terms of zero-shot and linear probe evaluation on various video tasks. Notably, our model even surpasses or is competitive with those trained on more data or modality, *e.g.*, InternVideo (Wang et al., 2022c) and ImageBind (Girdhar et al., 2023). In a nutshell, we make a step towards learning out-of-the-box spatiotemporal visual representation despite some limitations: (i) Though performance improvements on the benchmark, the emergent abilities are not present yet. (ii) The largest model studied in this paper is still far from the SOTA model in the image domain (*i.e.*, ViT-22B (Dehghani et al., 2023)).

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

## A    DOWNSTREAM TASKS

**Text-to-Video Retrieval.** The statistics of the three zero-shot text-to-video retrieval benchmarks are listed as follows: (a) **MSR-VTT** (Xu et al., 2016a) consists of 10K videos harvested from YouTube, and there are around 200K descriptions. We follow prior works (Luo et al., 2022; Xue et al., 2023; Wang et al., 2022c) to conduct evaluations on the 1K-A test set. (b) **DiDeMo** (Anne Hendricks et al., 2017) consists of 10K Flickr videos with around 40K sentences. Following Luo et al. (2022); Ge et al. (2022a); Bain et al. (2021), we concatenate all sentences that describe the same video to form a single query and conduct paragraph-to-video retrieval. Specifically, we do not crop and concatenate the localized moments but directly use the whole video in the retrieval set, as done by (Ge et al., 2022a; Bain et al., 2021). (c) **LSMDC** (Rohrbach et al., 2015) has 118,081 videos cropped from 202 movies. The evaluation protocol follows Zeng et al. (2023); Ge et al. (2022a), where the test set contains 1,000 videos.

**Action Recognition and Anomaly Detection.** The statistics of the six benchmarks are listed as follows: (a) **HMDB-51** (Kuehne et al., 2011) contains 5K videos of 51 action categories. The training and test sets have 3.5K and 1.5K videos, respectively. (b) **UCF-101** (Soomro et al., 2012) contains 13K videos of 101 action categories. The training set has 9.5K videos, and the test set has 3.5K videos. (c) **Kinetics-400** (Kay et al., 2017) is a large-scale dataset with 260K videos belonging to 400 categories, where 240K videos are used for training, and 20K videos are used for validation. (d) **Kinetics-600** (Carreira et al., 2018) has 480K videos belonging to 600 categories. The training, validation, and test sets have 390K, 30K, and 60K videos, respectively. (e) **SSV2** (Goyal et al., 2017) consists of 189K videos showing humans performing 174 pre-defined fine-grained actions with everyday objects. The training set has 169K videos, and the validation set contains 20K videos. (f) **UCF-Crime** (Sultani et al., 2018) consists of 1.9K long untrimmed surveillance videos that cover 13 real-world anomalies. We evaluate the performance on the official test split.

In zero-shot action recognition, we follow the prior work (Radford et al., 2021) to use the prompt template "a person [CLASS]" for **HMDB-51**, **UCF-101**, **Kinetics-400**, and **Kinetics-600**, where the cosine similarity is calculated between all video-category pairs for classification. As for **SSV2**, we turn it into a multi-choice task, namely **SSV2-MC**. For each video, we randomly pick 173 negative descriptions from other categories (one per category) and put them along with the ground-truth one into a candidate set. The model is expected to retrieve the right one from the 174 candidates. For **UCF-Crime**, we use the prompt template "a video of [EVENT]" to recognize the anomaly events in a zero-shot action recognition manner.

## B    IMPLEMENTATION DETAILS

We use AdamW (Loshchilov & Hutter, 2018) as the optimizer with a weight decay of $0.05$. The initial learning rates are set to be $1 \times 10^{-4}$ and $1 \times 10^{-7}$ for the newly added modules and the origin CLIP modules, respectively. We fix the weight parameter $\lambda$ in $\mathcal{L}$ to be 2 for roughly scaling the gradient magnitudes of $\mathcal{L}_{\text{VTC}}$ and $\mathcal{L}_{\text{TS}}$ to be the same. For the text encoder of all models, we freeze the first three-quarters of layers, leaving other layers trainable. Due to computation source limitation, we vary the batch size for different models, and all hyper-parameters are listed in Table 8. For our ViT-

Table 8: The hyper-parameters for pre-training and liner probe.

| config | pre-training | | | linear probe |
|---|---|---|---|---|
| | B/32 | B/16 | H/14 | all models |
| optimizer | | AdamW | | SGD |
| weight decay | | 0.05 | | 0 |
| training epochs | | 10 | 5 | 100 |
| learning rate | | $1 \times 10^{-4}$ (new modules) | | 0.1 |
| | | $1 \times 10^{-7}$ (CLIP modules) | | |
| batch size | 768 | 768 | 160 | 512 |
| frozen text layers | | 1-9 | 1-18 | n/a |
| input frames | | 12 | | 12 |
| masking ratio | 0 | 50 | 70 | 0 |
| augmentation | | RandomCrop | | CenterCrop |
| GPU for training | 32 V100 | 64 V100 | 80 V100 | 16 V100 |

B/32 and ViT-B/16 models, we load the weights released by OpenAI (Radford et al., 2021). As for the largest ViT-H/14 model, we inherit weights from OpenCLIP (Schuhmann et al., 2022). The detailed model architectures are listed in Table 9.

For masking ratio selection, according to our experiments, when keeping the training steps consistent, a larger ratio of masking could only degrade the performance rather than bring extra gains, as shown in Table 10. We select mask ratios according to the principle that achieving optimal performance with affordable resources. Note that we use the same mask ratio in all our ablation studies to rule out the potential impact of this factor and conduct fair comparisons.

Table 9: Our detailed model architectures. "Embed" and "Hidden" denote the dimension of the shared and hidden representations, respectively.

| Model | Params | Embed | GFLOPs | Video Encoder | | | | | Text Encoder | | | |
|-------|--------|-------|--------|-------|--------|--------|-------------|--------|--------|--------|--------|--------|
| | | | | Layers | Params | Hidden | Patch | GFLOPs | Layers | Params | Hidden | GFLOPs |
| B/32 | 149M | 512 | 72 | 12 | 86M | 768 | $32 \times 32$ | 69 | 12 | 63M | 512 | 3 |
| B/16 | 149M | 512 | 281 | 12 | 86M | 768 | $16 \times 16$ | 278 | 12 | 63M | 512 | 3 |
| H/14 | 1.0B | 1024 | 2674 | 32 | 632M | 1280 | $14 \times 14$ | 2650 | 24 | 354M | 1024 | 24 |

Table 10: The training configuration *w.r.t.* different masking ratios. We report zero-shot R@1 on MSR-VTT for reference. BS denotes the overall batch size. Each V100 has 32 GB memory.

| Method | Masking Ratio | BS | GPU BS | GPU Mem | Overall GPU | GFLOPs | MSR-VTT |
|--------|---------------|-----|--------|---------|-------------|--------|---------|
| CLIP-B/16 | n/a | n/a | n/a | n/a | n/a | 278 | 31.8 |
| Ours-B/16 | 50% | 768 | 12 | 27.5 GB | 64 V100 | 138 | 35.9 |
| | 60% | | 16 | 29.0 GB | 48 V100 | 110 | 35.7 |
| | 70% | | 24 | 31.0 GB | 32 V100 | 82 | 35.0 |
| | 80% | | 32 | 29.2 GB | 24 V100 | 55 | 34.5 |

## C DIVIDED SPACE-TIME ATTENTION

In this section, we describe the divided space-time attention in our video encoder detailedly. As illustrated in Figure 3, for the intra-frame tokens, *i.e.*, spatial-related tokens, we add the same spatial positional embeddings, and for tokens at the same position across different frames, *i.e.*, temporal-related tokens, we add the same temporal positional tokens. For each token, it first attends to the temporal-related tokens, then attend to the spatial-related tokens. Note that the [CLS] token is attended in both temporal and spatial self-attention.

## D ADDITIONAL EXPERIMENTS

**Pre-training Data Comparison.** We have listed the pre-trained datasets and their scales in Table 11 for a clear comparison. Generally speaking, models are expected to achieve better performance with an increasing number of training data given the same training recipe. However, the performance does not always improve since more data bring more noise. Moreover, video data with curated alt-text, *e.g.*, WebVid, is generally hard to scale up. Therefore, it is important to study scalable and robust training schemes (our motivation) that can learn from large-scale noisy video data with naturally associated text knowledge, *e.g.*, ASR transcripts.

**Computational and Memory Analysis.** We provide a detailed computational and memory analysis during both training and inference in Table 12. During the training phase, we optimized the batch sizes to fully utilize available GPU memory. Our analysis, especially the GFLOPs comparison between the video and text encoders, reveals that the most computationally intensive part is video encoding, particularly the spatiotemporal attention module. However, the use of an efficiently designed divided space-time attention mechanism ensures that even our largest model, *i.e.*, ViT-H/14, can be run on consumer-level GPUs like the RTX 3090, thanks to its relatively modest memory requirements during inference.

**Direct Evidence for Text Distortion and Catastrophic Forgetting.** We further provide direct evidence linked to the distortion and catastrophic forgetting in Table 14. We observe that: (i) $\mathbf{M}_{\text{All-Full}}$ fits the ASR data better than $\mathbf{M}_{\text{All-Partial}}$ given the smaller losses for epochs 1/3/5. (ii) However, $\mathbf{M}_{\text{All-Full}}$ captures worse semantics given the lower test accuracy in terms of linear text classification on Yahoo

Table 14: The top-1 accuracy (%) on Yahoo Answers and average training loss for different epochs (EP). The tests are done with the officially released ViT-B/32 models.

| Method | Top-1 Acc. | Loss EP1. | Loss EP3. | Loss EP5. |
|--------|-----------|-----------|-----------|-----------|
| CLIP | 57.61 (+0.00) | n/a | n/a | n/a |
| $\mathbf{M}_{\text{All-Full}}$ | 56.90 (-0.71) | 9.76 | 8.43 | 7.99 |
| $\mathbf{M}_{\text{All-Partial}}$ | 58.92 (+1.31) | 10.75 | 9.00 | 8.62 |

News (Zhang et al., 2015) (another language register than ASR). It indicates that the fully tuned $\mathbf{M}_{\text{All-Full}}$ indeed overfits to ASR styles and loses its generalization ability, as we claimed.

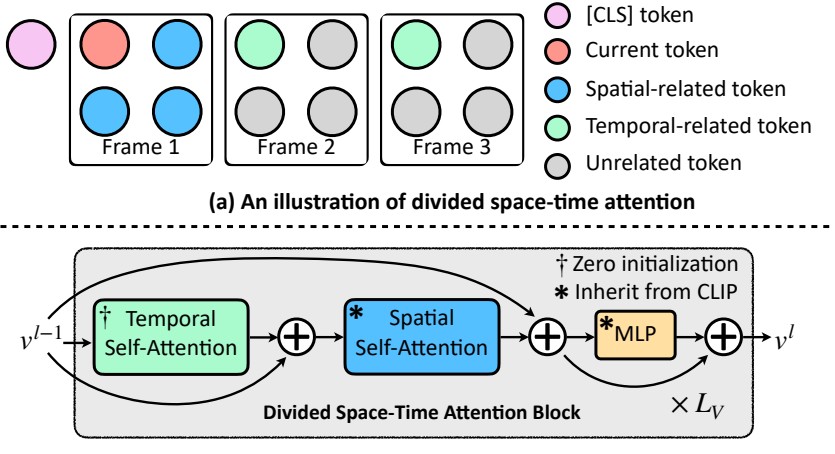

(a) An illustration of divided space-time attention

(b) The detailed structure of the divided space-time attention block

Figure 3: (a) We illustrate the divided space-time attention. For the orange token, it attends to tokens belonging to the same frame, *i.e.*, spatial-related tokens, and tokens in the same position across different frames, *i.e.*, temporal-related tokens, as well as the [CLS] token. (b) The structure of the divided space-time attention block, where we initialize the parameters of the temporal self-attention module with zeros and inherit weights from CLIP for the spatial self-attention and MLP.

**Linear Classification Results.** In addition, we provide the linear classification results in Table 13, where we optimize a linear classifier added on top of the frozen visual encoder. We surpass either self-supervised or language-guided models. Notably, we find the linear accuracy of Video-MAE (Tong et al., 2022) and VideoMAEv2 (Wang et al., 2023) lags far behind other models. It implies that representation learned by MAE-style objectives concentrates on the pixel level instead of the semantic level, making them impractical for learning general-purpose features. By contrast, with the aid of language supervision, our model is capable of out-of-the-box transferring and retains training efficiency like MAE variants.

**Fine-tuning Results.** While the primary objective of this paper is to promote out-of-the-box spatiotemporal representation learning, we also investigate whether this training framework negatively impacts the fine-tuning performance using the ViT-B/32 model. The results can be found in Table 15, where our TVTSv2 attains state-of-the-art results, suggesting that the proposed training approach plays a role in enhancing the fine-tuning performance to a certain degree.

Table 15: The fine-tuning performance on DiDeMo and LSMDC. All baselines are based on the ViT-B/32 model.

| Method | DiDeMo | | LSMDC | |
|---|---|---|---|---|
| | R@1 | R@10 | R@1 | R@10 |
| CLIP4Clip (Luo et al., 2022) | 43.4 | 80.6 | 21.6 | 49.8 |
| CenterCLIP (Zhao et al., 2022) | - | - | 21.7 | 49.8 |
| XPool (Gorti et al., 2022) | - | - | 22.7 | 51.2 |
| CLIP2TV (Gao et al., 2021) | 45.5 | 80.6 | - | - |
| Ours | **45.5** | **82.1** | **24.0** | **54.1** |

**Comparison between Adapter-wise Tuning and Partial Freezing on Text Encoder.** To underscore the importance of partial freezing, we introduce a ViT-B/32-based alternate baseline, integrating adapters (Houlsby et al., 2019) between text encoder layers, denoted as $M_{adapter}$. During training, only the adapter-introduced parameters are trainable. The MSR-VTT retrieval results, detailed in Table 16, show that this alternative method yields comparatively lower performance. This outcome reinforces our assertion that end-to-end tuning can cause style overfitting, even when the text encoder remains unaltered. It also confirms that our approach of partially frozen tuning is not just a superficial fix, but a substantive and effective solution to the identified problem.

Table 16: The zero-shot text-to-video retrieval results on MSR-VTT *w.r.t.* different textual training strategies. PF denotes partial freezing, and adapter denotes freezing the original parameters and plugging adapters between layers.

| Method | R@1 | R@5 | R@10 | MdR |
|---|---|---|---|---|
| CLIP | 30.6 | 54.4 | 64.3 | 4.0 |
| $M_{PF}$ | 34.5 | 58.5 | 67.7 | 3.5 |
| $M_{adapter}$ | 28.8 | 54.8 | 65.7 | 4.0 |

Table 11: The pre-training datasets of different methods. WV2.5M, K400, and K700 are short for WebVid-2.5M, Kinetics-400 and Kinetics-700, respectively. We only count the actual videos instead of the split clips. ∗ means accessing extra modalities, *e.g.*, audio. Single-stream models are de-emphasized. We report zero-shot R@1 on MSR-VTT for reference.

| Method | Pre-training Datasets | # Videos/Images | MSR-VTT |
|---|---|---|---|
| *Non-CLIP models* | | | |
| VideoCLIP | HowTo100M | 1.2M | 10.4 |
| Frozen | CC3M + WV2.5M | 5.5M | 18.7 |
| ALPRO | CC3M + WV2.5M | 5.5M | 24.1 |
| VIOLET | YT-Temporal + WV2.5M + CC3M | 11.5M | 25.9 |
| BridgeFormer | CC3M + WV2.5M | 5.5M | 26.0 |
| OmniVL | COCO + VG + CC3M + CC12M + SBU + WV2.5M + ImageNet-1K + K400 | ∼16.5M | 34.6 |
| *CLIP-B/32* | | | |
| CLIP-B/32 | WIT | 400M | 30.6 |
| CLIP-straight | WIT | 400M | 31.2 |
| CLIP4Clip | WIT + HowTo100M-380k | 400M + 380K | 32.0 |
| BridgeFormer | WIT + CC3M + WV2.5M | 400M + 5.5M | 33.2 |
| CLIP-ViP | WIT + HD-VILA-100M | 400M + 3.3M | 29.0 |
| Ours-B/32 | WIT + YT-Temporal + WV2.5M | 400M + 7.5M | 34.5 |
| *CLIP-B/16* | | | |
| CLIP-B/16 | WIT | 400M | 31.8 |
| CLIP-ViP | WIT + YT-Temporal + WV2.5M | 400M + 3.3M | 31.7 |
| UMT-B | WIT + K700 + CC3M + WV2.5M | 400M + 6.15M | 29.6 |
| Ours-B/16 | WIT + YT-Temporal + WV2.5M | 400M + 7.5M | 35.9 |
| *Larger Models* | | | |
| ImageBind∗ | LAION-400M + Various Datasets | 400M + ? | 36.8 |
| InternVideo | WIT + LAION-100M + WebVid2.5M + WebVid10M + HowTo100M | 400M + 113.7M | 40.0 |
| UML-L | WIT + K700 + CC3M + WV2.5M | 400M + 6.15M | 33.3 |
| Ours-H/14 | LAION-400M + YT-Temporal + WV2.5M | 400M + 7.5M | 38.2 |

Table 12: The computational and memory analysis during training and inference. "BS" denotes the batch size. V-GLOPs and T-GLOPs refer to GFLOPs corresponding to the video and text encoder.

| Model | Training BS × GPU | Training Mem | Inference Mem | V-GFLOPs | T-GFLOPs |
|---|---|---|---|---|---|
| B/32 | 24 × 32 | 30.2 GB | 3.3 GB | 69 | 3 |
| B/16 | 12 × 64 | 27.5 GB | 4.5 GB | 278 | 3 |
| H/14 | 2 × 80 | 31.3 GB | 8.8 GB | 2650 | 24 |

**The Partial Freezing Effects on Video Encoder.** Since it is natural to question if video distortion happens in the visual branch, we conduct such experiments and report results on MSR-VTT in Table 17. We consider two variants: (**i**) $M_{visual\ PF}$ that partially freezes the image encoder. (**ii**) $M_{AIM}$ that embraces a parameter-efficient fine-tuning method, *i.e.*, AIM (Yang et al., 2023), in which adapters are inserted into the frozen image encoder. All models adopt ViT-B/32 as the backbone, and the text encoder is partially frozen.

Table 17: The zero-shot text-to-video retrieval results on MSR-VTT *w.r.t.* different visual training strategies. "FT" denotes full tune, "PF" denotes partial freezing, and "AIM" denotes plugging AIM adapters between frozen layers.

| Method | R@1 | R@5 | R@10 | MdR |
|---|---|---|---|---|
| $M_{visual\ FT}$ | 34.5 | 58.5 | 67.7 | 3.5 |
| $M_{visual\ PF}$ | 32.5 | 56.3 | 65.9 | 4.0 |
| $M_{visual\ AIM}$ | 31.0 | 54.4 | 66.3 | 4.0 |

We observe that partially freezing visual models does not help. The performance is inferior to the original one with a fully tuned visual encoder. It indicates that the visual branch does not suffer from "video distortion", and for pre-training with large-scale data, it is beneficial to leave all visual parameters trainable to better capture temporal knowledge and pursue stronger representations.

Table 13: The **linear action recognition** results of top-1 accuracy. We de-emphasize DI-NOv2 (Oquab et al., 2023) because it uses a larger model than ours, *i.e.*, ViT-g/14. "Our impl." denotes we use official pre-trained weights for evaluation.

| Method | Venue | Cite From | Supervision | Params | HMDB-51 | UCF-101 | Kinetics-400 |
|---|---|---|---|---|---|---|---|
| MemDPC (Han et al., 2019) | ECCV'20 | SVT | | | 30.5 | 54.1 | - |
| CoCLR (Han et al., 2020) | NeurIPS'20 | SVT | | | 52.4 | 77.8 | - |
| Vi²CLR (Diba et al., 2021) | ICCV'21 | SVT | | | 47.3 | 75.4 | 63.4 |
| VideoMoCo (Pan et al., 2021) | CVPR'21 | SVT | | | 49.2 | 78.7 | - |
| CVRL (Qian et al., 2021) | CVPR'21 | SVT | Self | < 100M | 57.3 | 89.2 | 67.6 |
| DINO (Caron et al., 2021) | ICCV'21 | DINOv2 | | | - | 85.0 | 64.5 |
| SVT (Ranasinghe et al., 2022) | CVPR'22 | SVT | | | 57.8 | 90.8 | 68.1 |
| iBOT (Zhou et al., 2022) | ICLR'22 | DINOv2 | | | - | 88.6 | 72.6 |
| VideoMAE-B (Tong et al., 2022) | NeurIPS'22 | Our impl. | | | 30.9 | 52.7 | 20.4 |
| DINOv2-g (Oquab et al., 2023) | arXiv'23 | DINOv2 | | 1.0B | - | 91.2 | 78.4 |
| VideoMAEv2-H (Wang et al., 2023) | CVPR'23 | Our impl. | | 632M | 34.1 | 56.4 | 25.8 |
| CLIP-B/16 (Radford et al., 2021) | ICML'21 | Our impl. | | | 62.8 | 87.6 | 66.9 |
| Frozen (Bain et al., 2021) | ICCV'21 | Our impl. | | | 57.8 | 88.7 | 62.9 |
| MERLOT (Zellers et al., 2021) | NeurIPS'21 | MERLOT | Language | < 100M | 49.6 | 74.9 | - |
| BridgeFormer (Ge et al., 2022a) | CVPR'22 | Our impl. | | | 60.7 | 89.2 | 65.6 |
| MILES (Ge et al., 2022b) | CVPR'22 | Our impl. | | | 60.0 | 89.6 | 64.0 |
| TVTS (Zeng et al., 2023) | CVPR'23 | Our impl. | | | 60.1 | 87.6 | 60.8 |
| Ours-B/16 | - | - | | | 64.7 | 90.0 | 70.1 |
| Ours-H/14 | - | - | | 632M | **65.7** | **91.8** | **73.1** |

**Robustness Evaluation.** Since robustness plays an important role in developing foundation models, we further test the performance under noisy or incomplete data by randomly masking some patches in each frame during inference. The retrieval performance under different masking ratios on MSR-VTT is reported in Table 18. Thanks to the masked contrastive pre-training, our models are robust to data corruption. Impressively, even minor data distortions, such as 10% or 20% masking, do not significantly impact performance. More notably, our models demonstrate commendable robustness, maintaining considerable effectiveness even under substantial data incompleteness, with a masking ratio as high as 50%. This clearly underscores the reliability and adaptability of our models in handling corrupted data scenarios.

Table 18: The performance under corrupted data on MSR-VTT.

| Model | Masking Ratio | R@1 | R@5 | R@10 | MdR |
|---|---|---|---|---|---|
| B/16 | 0 | 35.9 | 61.2 | 71.3 | 3.0 |
| | 10 | 35.4 | 61.1 | 71.0 | 3.0 |
| | 20 | 35.0 | 60.6 | 70.4 | 3.0 |
| | 50 | 32.6 | 60.0 | 69.9 | 3.0 |
| H/14 | 0 | 38.2 | 62.4 | 73.2 | 3.0 |
| | 10 | 37.3 | 62.1 | 72.7 | 3.0 |
| | 20 | 36.0 | 61.6 | 72.1 | 3.0 |
| | 50 | 35.2 | 60.4 | 70.1 | 3.0 |

**Temporal Module Initialization.** We analyze the proper way to instantiate temporal modules based on the ViT-B/32 model. Since both the attention architecture and initialization weights are essential to the final performance, we further supplement a baseline of joint attention. As shown in Table 19, joint attention performs worst because the simple extension of attention range hurts the spatial prior. Besides, initializing temporal attention weights with zeros beats its random initialization competitor by a large margin, indicating that growing temporal reasoning ability out of nothing avoids hurting the well-learned spatial prior, thus more suitable for training video foundation models.

Table 19: The **zero-shot text-to-video retrieval** results of R@1 *w.r.t.* different attention mechanisms and initialization strategies.

| Attention | Initialization | MSR-VTT | DiDeMo |
|---|---|---|---|
| Joint | CLIP | 30.8 | 28.8 |
| Divided Space-Time | S: CLIP T: random | 31.2 | 29.3 |
| Divided Space-Time | S: CLIP T: zero | 34.5 | 31.2 |

**Comparison with TVTSv1.** We further compare with TVTS (Zeng et al., 2023), which focuses on linear and fine-tuning tasks and adopts a weak initialization. As shown in Table, TVTS is not good at cross-modal retrieval and our TVTSv2 demonstrates great superiority.

Table 20: The zero-shot retrieval results *w.r.t.* TVTS and TVTSv2.

| Method | MSR-VTT | | | DiDeMo | | | LSMDC | | |
|---|---|---|---|---|---|---|---|---|---|
| | R@1 | R@5 | MdR | R@1 | R@5 | MdR | R@1 | R@5 | MdR |
| TVTS-B/16 | 18.1 | 38.5 | 12.0 | 18.6 | 39.0 | 10.0 | 8.6 | 21.5 | 55.0 |
| TVTSv2-B/32 | 34.5 | 58.5 | 3.5 | 31.2 | 56.9 | 4.0 | 16.1 | 30.6 | 25.0 |
| TVTSv2-B/16 | 35.9 | 61.2 | 3.0 | 33.4 | 60.1 | 3.0 | 16.9 | 31.5 | 22.0 |
| TVTSv2-H/14 | 41.3 | 63.0 | 2.0 | 39.5 | 63.6 | 3.0 | 20.0 | 37.8 | 11.0 |

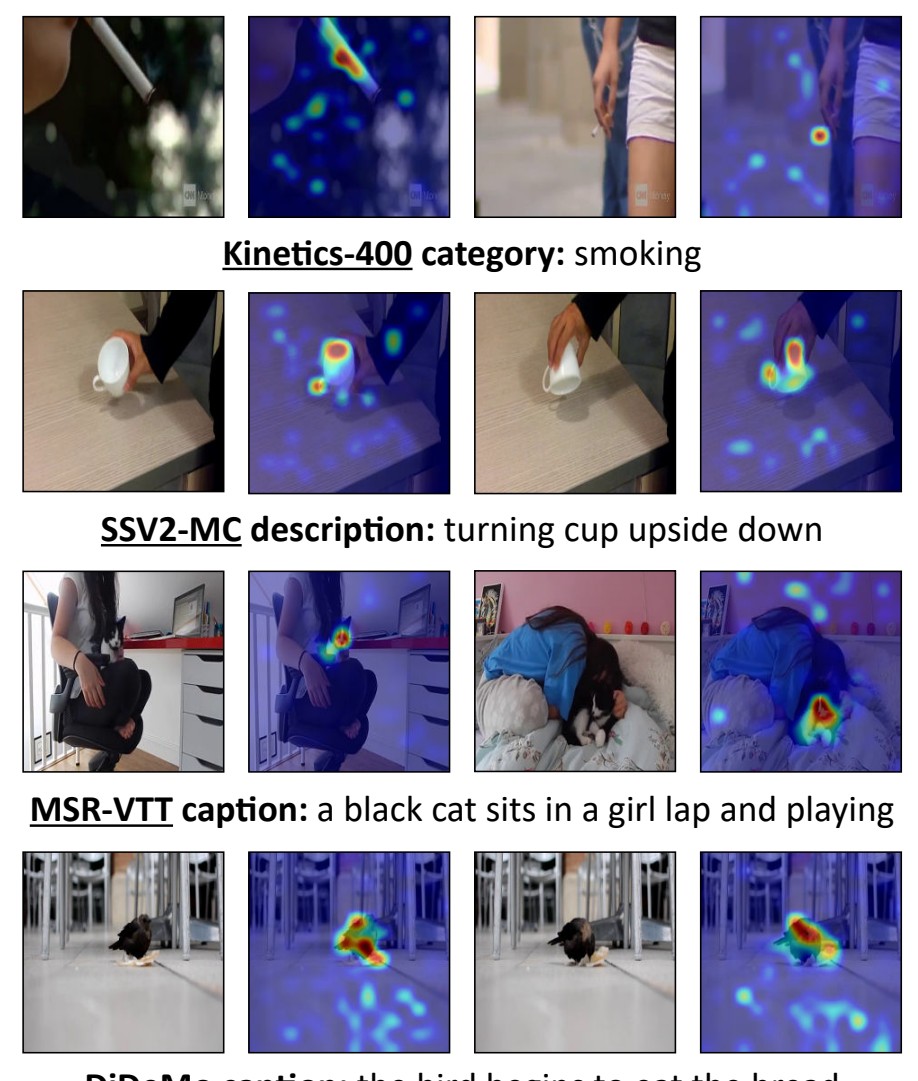

**Kinetics-400 category:** smoking

**SSV2-MC description:** turning cup upside down

**MSR-VTT caption:** a black cat sits in a girl lap and playing

**DiDeMo caption:** the bird begins to eat the bread

Figure 4: The self-attention map of the visual [CLS] token. *Note that it does not involve cross-modal alignment.* The captions are for reference only and are not used for attention maps. All crucial objects involved in the temporal interaction are captured precisely, which indicates our model's powerful ability to produce general-purpose features.

**Attention Visualization.** We visualize the zero-shot self-attention map of the visual [CLS] token on different datasets in Figure 4. The critical objects involved in the temporal interactions are captured precisely, indicating our model's strong out-of-the-box ability intuitively.

## E  BROADER IMPACT

The negative social impacts of our model may lie in intensifying global warming because of the large amount of carbon emission produced by GPU clusters. Though the pre-training phase is energy-consuming, the model can be used out-of-the-box, saving the potential carbon emission in fine-tuning. Since the model produces general-purpose representations, it might also raise the risk of abuse, such as unauthorized biometric recognition. In addition, it is our duty to guarantee there is little bias when releasing the foundation models. We have checked the accuracies corresponding to different sexes/races are comparable, indicating there is little risk of model bias.

