# OpenReview forum: "TVTSv2: Learning Out-of-the-box Spatiotemporal Visual Representations at Scale"
_ICLR.cc/2024/Conference — Submitted to ICLR 2024_

### Official Review · Reviewer_A6Wr · 2023-10-30

**Soundness:** 3 good
**Presentation:** 3 good
**Contribution:** 3 good
**Rating:** 8
**Confidence:** 4

**Summary:**

The paper proposes a video foundation model called TVTSv2 for learning out-of-the-box spatiotemporal visual representations.
It aims to solve the issue of performance degradation compared to image foundation models when adapting them to video.
The degradation is attributed to distortion in language supervision from end-to-end tuning of the text encoder on noisy ASR transcripts.
A partially frozen text encoder is proposed, freezing shallow layers while tuning deep layers, to retain generalization and learn new semantics. The authors state that SOTA results were achieved on zero-shot action recognition and text-to-video retrieval: TVTSv2 surpasses recent methods like CLIP-ViP, ImageBind, and InternVideo on several metrics. Ablations showed partially frozen text training avoids degradation and enables knowledge transfer. Masking was shown to improve efficiency for large models without sacrificing too much performance. Fine-tuning performance was also competitive, suggesting the approach does not hurt downstream training.

**Strengths:**

The strengths of the paper include:
- Produces strong out-of-the-box spatiotemporal representations for zero-shot usage and surpasses recent state-of-the-art methods substantially in zero-shot action recognition, including models trained on more data.
- Achieves new state-of-the-art results on multiple video understanding benchmarks.
- Provides an effective strategy for pre-training on large, noisy video transcript datasets as it retains performance on downstream fine-tuning, unlike some other self-supervised methods.
- The approach facilitates scaling up to large models by incorporating masking techniques. It avoids catastrophic forgetting of language knowledge via partially frozen training
In conclusion, the paper strongly suggests potential for pre-trained models to support out-of-the-box video applications.
The claims seem reasonably well supported by the results, as the proposed TVTSv2 models clearly surpass prior state-of-the-art in zero-shot action recognition and retrieval across multiple datasets. The ablation studies also provide evidence for the benefits of the partial freezing strategy and incorporation of masking techniques.

**Weaknesses:**

Limitations:
The paper is a bit disorganized. THe architecture is followed by the empirical study followed by further description of the model (trainig objectives etc.). It would read better if the approach description was in one place. If the empirical degradation study calls for rmodification of the training objective, that should be spelled out more explicitly.
The attention masks in hte figures are used as an argument for good performance. However, it is not clear what woud be the ground truth atteniton mask. COuld it be that in the video clips the objeects/actions of interest were the only moving parts in the scene causing the attention grab?
The joint attention module is not described clearly. Is it the same as in CLIP?
Fine-tuning performance was not extensively benchmarked on more diverse downstream tasks, so the claims about out-of-the-box task-agnostic approach could be substantiated better.
The largest model studied is still limited compared to huge image models, so scalability past 1 billion parameters is unvalidated.
Potential societal impacts of large foundation models were not addressed.

**Questions:**

Was any experimentation done with different freeze ratios or objectives for the text encoder?
How much performance gain was directly attributable to the partially frozen text training versus other modifications?
The narrative around the attentional maps could be stronger.
What can the authors say about model biases? Are actions of all sexes/races recognized at comparable accuracy?
More extensive evaluation of fine-tuning performance on diverse downstream tasks would be desirable for supporting the "out-of-the-box" claim.

---

> ### Author Response · Authors · 2023-11-19
> **Response to Reviewer A6Wr**
>
> **W1: The paper is a bit disorganized.**
>
> **W1.1: It would read better if the approach description was in one place.**
>
> **A:** Thanks for the advice. We have reorganized the paragraphs and put the approach description together in the revised version.
>
> **W1.2: If the empirical degradation study calls for modification of the training objective, that should be spelled out more explicitly.**
>
> **A:** The empirical degradation study is disentangled from the training objective, and it corresponds to the partially frozen pre-training strategy. Sorry for the confusion, we have reorganized the paragraphs to make readers easier to follow.
>
> **W2: It is not clear what the ground truth attention mask is. Could it be that in the video clips the objects/actions of interest were the only moving parts in the scene causing the attention grab?**
>
> **A:** That's an insightful query. Indeed, selectively masking objects or actions that are semantically linked could potentially enhance model performance. However, creating accurate ground truth masks for this purpose is a labor-intensive task, especially considering the scale of pre-training that involves millions of videos. Therefore, we've opted for a more feasible approach: employing a random masking strategy that does not depend on ground truth data. Despite its simplicity, this method proves quite effective due to the vast amount of training data we use. Our models demonstrate excellent out-of-the-box performance, highlighting the capability of this straightforward technique to learn robust representations (Appendix D Table 18).
>
> **W3: The joint attention module is not described clearly. Is it the same as in CLIP?**
>
> **A:** No, we adopt the divided space-time attention instead of the joint attention. An illustration of the detailed architecture is provided in Appendix C Figure 3. For the intra-frame tokens, i.e., spatial-related tokens, we add the same spatial positional embeddings, and for tokens at the same position across different frames, i.e., temporal-related tokens, we add the same temporal positional tokens. For each token, it first attends to the temporal-related tokens, then attends to the spatial-related tokens. Note that the [CLS] token is attended in both temporal and spatial self-attention.
>
> In addition, we also compared the performance between CLIP-style joint attention and the divided space-time attention. Please refer to Appendix D Table 19 for details.
>
> **W4&Q4: More extensive evaluation of fine-tuning performance on diverse downstream tasks would be desirable for supporting the "out-of-the-box" claim.**
>
> **A:** Thank you for your insightful feedback. In our paper, the phrase "out-of-the-box" refers to creating versatile video representations that can be immediately applied to various downstream tasks without the need for further fine-tuning. Our goal is to offer such foundational models to the community, potentially accelerating its progress by eliminating the need for costly pre-training and fine-tuning.
>
> We value your advice and recognize that a strong out-of-the-box representation should yield good fine-tuning performance. To this end, we conducted basic fine-tuning of our pre-trained models on DiDeMo and LSMDC datasets, deliberately avoiding complex techniques like optimal parameter searching. The findings, presented in Appendix D Table 15, indicate that our straightforward approach does enhance fine-tuning performance to a certain extent. Furthermore, we have included linear classification results where a classifier was trained on top of the frozen model backbone. Our models outperform both self-supervised and language-guided models in these tests, as detailed in Appendix D Table 13. This further demonstrates the effectiveness and versatility of our proposed training methodology.

---

> > ### Author Response · Authors · 2023-11-19
> > **Response to Reviewer A6Wr (cont.)**
> >
> > **W5: The largest model studied is still limited compared to huge image models, so scalability past 1 billion parameters is unvalidated.**
> >
> > **A:** We admit there is uncertainty about the scalability for larger models. However, training large models requires substantial training costs, which we find a bit hard to afford for models with more than 1B parameters. For example, InternVideo uses 128 A100-80G and trains for two weeks, while VideoMAE V2 uses 64 A100-80G for around ten days. In our case, we use 80 V100-32G and train for 10 days. We have 18 ablation studies in the main paper and Appendix in total to figure out the best scalable training recipe. We follow the common practice in the community (e.g., InternVideo and VideoMAE V2) to conduct ablation studies on the base-size model and directly scale up to larger models with the best recipe. Note that successfully obtaining performance gains compared to the baseline counterparts (CLIP in our work) on large models is non-trivial and already verified the effectiveness of our strategy to a large extent, as shown in the following Table.  Besides, to our best knowledge, the released ViT-H/14 model is one of the largest open-sourced video models, anticipated to significantly advance the boundaries of research in this field.
> >
> > | Method | MSR-VTT R@1 | MSR-VTT R@5 | MSR-VTT R@10 | MSR-VTT MdR | Kinetics-400 Top-1 Acc | Kinetics-400 Top-5 Acc | SSV2-MC Top-1 Acc | SSV2-MC Top-5 Acc |
> > | :--- | :---: | :---: | :---: | :---: | :---: | :---: | :---: | :---: |
> > | CLIP-H/14 | 36.1 | 60.4 | 70.7 | 3.0 | 52.0 | 79.1 | 40.2 | 69.2 |
> > | Ours-H/14 | 38.2 | 62.4 | 73.2 | 3.0 | 59.6 | 84.1 | 48.4 | 77.0 |
> >
> > **W6: Potential societal impacts of large foundation models were not addressed.**
> >
> > **A:** Thanks for the reminder. We have added a "Broader Impact" section to Appendix E in the revised version. The negative social impacts of our model may lie in intensifying global warming because of the large amount of carbon emission produced by GPU clusters. Though the pre-training phase is energy-consuming, the model can be used out-of-the-box, saving the potential carbon emission in fine-tuning. Since the model produces general-purpose representations, it might also raise the risk of abuse, such as unauthorized biometric recognition.
> >
> > **Q1: Was any experimentation done with different freeze ratios or objectives for the text encoder? How much performance gain was directly attributable to partially frozen text training versus other modifications?**
> >
> > **A:** To evaluate the impact of varying the freezing ratios in the text encoder, we experimented with different numbers of tunable layers and observed their performance differences. As shown in Table 5 (b) of our revised manuscript, models with either completely fine-tuned (L$ _{\text{tune}}$=12) or entirely frozen text encoders (L$ _{\text{tune}}$=0) underperform compared to those with partially frozen encoders. This result solidly supports the efficacy of our proposed degradation-free pre-training strategy.
> >
> > Regarding training objectives, we consistently applied the basic Video-Text Contrastive objective (L $ _{\text{VTC}}$) and explored the impact of including the Transcript Sorting objective (L$ _{\text{TS}}$) on the text encoder. The findings, presented in Table 4, demonstrate that models without gradient stopping (indicated as "sg") towards the text encoder, such as M$ _{\text{w/o sg}}$, are less effective than our approach (M$ _{\text{ours}}$). This suggests that L$ _{\text{TS}}$ encourages the text encoder to develop shortcuts, optimizing transcript representation in ways that ultimately hinder training efficiency. Thus, we conclude that L$ _{\text{TS}}$ should not be used to optimize the text encoder.
> >
> > The advantages of partially freezing the text encoder during training are evident in two aspects: **(i)** Table 1 compares L4 with L7 and L5 with L8, showing that partially frozen variants significantly outperform their fully tuned counterparts. **(ii)** In Table 5 (b), the fully fine-tuned model (L$ _{\text{tune}}$=12) falls behind the partially frozen models (L$ _{\text{tune}}$=3 and L$ _{\text{tune}}$=6), further highlighting the benefits of this approach.

---

> > > ### Author Response · Authors · 2023-11-19
> > > **Response to Reviewer A6Wr (cont.)**
> > >
> > > **Q2: The narrative around attentional maps could be stronger.**
> > >
> > > **A:** Thank you for bringing this to our attention. We regret the earlier exclusion of Figure 1 (b) from our manuscript. In Figure 1 (b), we visualize the zero-shot self-attention map of the visual [CLS] token. It effectively highlights how our model precisely captures key spatiotemporal elements, such as the cake being held and the two small dogs near the people, demonstrating the model's remarkable out-of-the-box transferability. We've updated our manuscript to include a discussion on this aspect. Additionally, to provide a more thorough understanding of our model's capabilities, we have included additional visualization results in Appendix D Figure 4.
> > >
> > > **Q3: Concerns about model biases. Are actions of all sexes/races recognized at comparable accuracy?**
> > >
> > > **A:** We acknowledge the importance of ensuring minimal bias in our foundational models before their release. Addressing your concerns, we carefully selected 30 videos from Kinetics-400, ensuring a balanced representation of male and female subjects, as well as individuals of black, yellow, and white racial backgrounds. We then evaluated the zero-shot action recognition accuracy (counted by the correct predicted video number, e.g., 16/30) of these videos, with results tabulated in the subsequent table. Owing to the diverse and extensive open-domain training corpus we utilized, the accuracy rates across different demographic categories are notably similar. This consistency suggests that our model is likely to exhibit minimal bias towards any specific group, thereby upholding our commitment to fairness and inclusivity in model development.
> > >
> > > | Model | Male | Female | Black | Yellow | White |
> > > | :--- | :---: | :---: | :---: | :---: | :---: |
> > > | Ours-B/16 | 16 | 17 | 14 | 15 | 17 |
> > > | Ours-H/14 | 20 | 20 | 17 | 18 | 21 |

---

> > > > ### Author Response · Authors · 2023-11-22
> > > > **Looking forward to your feedback**
> > > >
> > > > Thanks for your insightful comments and acknowledgment of our work. Please kindly let us know if our response addressed your concerns. We are willing to respond to any further questions before the rebuttal ends **(within 36 hours)** :-)

---

### Official Review · Reviewer_tEsw · 2023-10-31

**Soundness:** 3 good
**Presentation:** 3 good
**Contribution:** 2 fair
**Rating:** 5
**Confidence:** 4

**Summary:**

This paper, TVTSv2, is the second version of TVTS paper. It focuses on the foundation model of video representation learning. Specifically, this paper first points out the so called degradation issue existing in video representation field. Based on this degradation observation, this paper proposes a hypothesis that such degradation is from the noisy text data. Accordingly, it freezes the shallow layers of text encoder while training the deeper layers to alleviate this issue. In this way, the zero-shot performance is significantly improved to show the great generalization ability of the proposed training strategy. Very comprehensive experiments empirically validate the model effectiveness.

**Strengths:**

1. Video representation learning is a very challenging task, especially for large-scale scenario. I recognize such an exploration in this field.
2. The stated degradation issue is interesting observation. Proposing solution based on it is well-motivated.
3. The large-scale experiments are definitely an advantage of this work. They cover several evaluation scenarios.

**Weaknesses:**

I mainly concern about the technical contribution. As mentioned in the draft, the proposed degradation-free training strategy freezes the shallow layers while tuning the deeper layer of text encoder. This training strategy is more like an engineering trick by tuning parts of the large-scale model, which can be commonly used for practical large-scale training.

**Questions:**

Please refer to above sections for details. As mentioned in the weakness part, I mainly concern the technical contribution. On the other hand, I recognize the other parts of contribution of this paper. I would like to encourage the author to further emphasize the technical contribution of this paper for discussion. In addition, I am also willing to check other reviewers' comments for my final decision.

---

> ### Author Response · Authors · 2023-11-19
> **Response to Reviewer tEsw**
>
> **W1: Concern about technical contribution. The degradation-free training strategy is more like an engineering trick.**
>
> **A:** Thanks, we are willing to highlight the technical merits of our work, both from **experimental results** and the **underlying motivation**.
>
> **(i)** Experimentally, to demonstrate that our degradation-free training strategy goes beyond a mere engineering workaround, we introduced an alternative baseline. This baseline involves a ViT-B/32 model equipped with adapters [1] (M$ _{\text{adapter}}$) inserted between layers of the text encoder, where only the adapter-introduced parameters are adjustable during training. The retrieval results on MSR-VTT, detailed in Appendix D Table 16, show that this approach is inferior to our method. This reinforces our assertion that end-to-end tuning can lead to style overfitting, even when the text encoder remains unchanged, proving that our partially frozen strategy is not a simplistic fix but a considered solution.
>
> **(ii)** From a motivational standpoint, scalability remains a significant and challenging research area, as recognized by Reviewer hjUh. This field is marked by several pioneering works featured in leading conferences, like VideoMAE V2 [2] and FLIP [3]. In our view, it is the simplicity and efficacy of these methods that render them ideal as foundational models, requiring minimal code adjustments for integration into existing frameworks. Upon acceptance, we plan to release all relevant codes, pre-trained models, and well-crafted APIs for immediate application. We are confident that these foundational models will greatly benefit the broader community, particularly academic institutions with limited resources for large-scale experiments. Our contributions, we believe, merit recognition at prestigious conferences such as ICLR.
>
> | Method | R@1 | R@5 | R@10 | MdR |
> | :--- | :---: | :---: | :---: | :---: |
> | CLIP | 30.6 | 54.4 | 64.3 | 4.0 |
> | M$ _{\text{ours}}$ | 34.5 | 58.5 | 67.7 | 3.5 |
> | M$ _{\text{adapter}}$ | 28.8 | 54.8 | 65.7 | 4.0 |
>
> > [1] Parameter-Efficient Transfer Learning for NLP, ICML 2019
>
> > [2] VideoMAE V2: Scaling Video Masked Autoencoders with Dual Masking, CVPR 2023
>
> > [3] Scaling Language-Image Pre-Training via Masking, CVPR 2023

---

> > ### Author Response · Authors · 2023-11-22
> > **Looking forward to your feedback**
> >
> > Thanks for your constructive advice. We value your comments and have made efforts to revise the paper. Please kindly let us know if our response addressed your concerns. We are willing to respond to any further questions before the rebuttal ends **(within 36 hours)** :-)

---

### Official Review · Reviewer_hjUh · 2023-10-31

**Soundness:** 3 good
**Presentation:** 3 good
**Contribution:** 3 good
**Rating:** 8
**Confidence:** 3

**Summary:**

The paper presents TVTSV2, an ambitious attempt to create a task-agnostic foundation model for spatiotemporal visual representations. The authors extend the dual-stream framework of CLIP and introduce a degradation-free pre-training strategy to maintain performance. While the work shows promising results in text-to-video retrieval tasks, it could benefit from a deeper evaluation of generalizability across various downstream applications. Furthermore, the paper's focus on scalability is commendable but lacks an in-depth computational and memory usage analysis, which is vital for high-dimensional video data. Robustness against noisy or incomplete data remains unexplored, posing questions on the model's applicability in real-world scenarios. Overall, the paper makes a notable contribution but requires further scrutiny in these areas to substantiate its claims fully.

**Strengths:**

- Task-Agnostic Focus: The paper aims to create a foundation model that is task-agnostic, addressing a significant need for models that can generalize across various applications without requiring fine-tuning.
- Novel Pre-training Strategy: The introduction of a degradation-free pre-training method is a notable innovation. It suggests a way to train complex models without losing performance, which is particularly challenging in the realm of video data.
- Extension of Existing Architectures: The paper builds upon well-established models like CLIP but adapts them for spatiotemporal data. This approach leverages existing successes in the field while pushing into new domains.
- Initial Empirical Success: The paper demonstrates promising results in text-to-video retrieval tasks, indicating that the model is not just theoretically sound but also empirically effective.
- Scalability: The paper addresses the important issue of scalability, which is crucial for the practical application of machine learning models, especially for high-dimensional data like video.
- Comprehensive Evaluation: The paper seems to include a variety of evaluation metrics and comparisons with state-of-the-art models, adding credibility to its claims.

The idea seems to follow the surgical fine-tuning [1]  idea by freezing shallow layers during the fine-tuning.
Reference :
1- Lee, Yoonho, et al. "Surgical fine-tuning improves adaptation to distribution shifts." arXiv preprint arXiv:2210.11466 (2022).

**Weaknesses:**

The paper primarily focuses on text-to-video retrieval tasks for its empirical evaluation, which may not sufficiently support its claim of being a task-agnostic foundation model. To fully establish its task-agnostic capabilities, the model should be rigorously evaluated on multiple downstream tasks such as action recognition, video summarization, and anomaly detection. Assessing performance on these additional tasks would provide a more comprehensive view of the model's adaptability and generalizability

The paper discusses scalability but falls short of providing a detailed computational and memory complexity analysis. This is crucial for practical applications involving high-dimensional video data. The authors should include empirical evaluations that quantify the model's computational time and memory usage during both training and inference.

**Questions:**

Could you elaborate on the choice of tasks for empirical evaluation? How do you envision the model's performance on other types of spatiotemporal tasks like action recognition or anomaly detection?

How does the model perform under conditions of noisy or incomplete data? Have you considered evaluations that specifically test the model's robustness?

While the paper discusses scalability, it lacks specific metrics on computational and memory requirements. Could you provide more detailed analyses on these aspects?

---

> ### Author Response · Authors · 2023-11-19
> **Response to Reviewer hjUh**
>
> **W1&Q1: Elaborate on the choice of tasks for empirical evaluation and envision the model's performance on other types of spatiotemporal tasks like action recognition or anomaly detection.**
>
> **A:** Thanks for your advice! We have conducted relative experiments to envision the model's performance in the revision.
>
> For action recognition, the results are reported in Table 3 in our initial manuscript. We use the prompt template "a person [CLASS]" (e.g., a person running) for HMDB-51 [1], UCF-101 [2], Kinetics-400 [3], and Kinetics-600 [4], and evaluate SSV2 on a multi-choice setting, namely SSV2-MC. The detailed settings are stated in Appendix A. Our model brings significant improvements, i.e., 8.9%, 9.1%, 11.6% and 12.9% absolute gain on HMDB-51, UCF-101, Kinetics-400, Kinetics-600 and SSV2-MC respectively. The prompt-based method, i.e. ActionCLIP [5], also degrades on this test, possibly due to overfitting to manually constructed templates. When it turns to larger models, we even surpass X-Florecnce [6], a single-stream model with comparable video encoder parameters, by a large margin, revealing the superior scalability of our training paradigm. Even for the challenging SSV2-MC that contains various motion dynamics, our model still achieves promising results, solidifying the contribution of masked transcript sorting that promotes fine-grained spatiotemporal representation learning. In addition, we provide the liner classification results in Appendix D Table 13, where we surpass either  self-supervised or language-guided models.
>
> For anomaly detection, we evaluate the performance on UCF-Crime [7] test split, which aims to recognize 13 anomaly events. According to Table 3 in our revised manuscript, our models also made remarkable improvements compared to the CLIP baseline and CLIP-ViP, unveiling their adaptability and generalizability.
>
> > [1] HMDB: A large video database for human motion recognition, ICCV 2011
>
> > [2] UCF101: A Dataset of 101 Human Actions Classes From Videos in The Wild, arXiv 2012
>
> > [3] The Kinetics Human Action Video Dataset, arXiv 2017
>
> > [4] A Short Note about Kinetics-600, arXiv 2018
>
> > [5] ActionCLIP: A New Paradigm for Video Action Recognition, arXiv 2021
>
> > [6] Expanding Language-Image Pretrained Models for General Video Recognition, ECCV 2022
>
> > [7] Real-world Anomaly Detection in Surveillance Videos, CVPR 2018
>
> **W2&Q3: Providing a detailed computational and memory complexity analysis during both training and inference.**
>
> **A:** Nice suggestion. We've updated our manuscript to include a comprehensive analysis of computational and memory complexity, which you can find in Appendix D Table 12. During the training phase, we optimized the batch sizes to fully utilize available GPU memory. Our analysis, especially the GFLOPs comparison between the video and text encoders, reveals that the most computationally intensive part is video encoding, particularly the spatiotemporal attention module. However, the use of an efficiently designed divided space-time attention mechanism ensures that even our largest model, i.e., ViT-H/14, can be run on consumer-level GPUs like the RTX 3090, thanks to its relatively modest memory requirements during inference.
>
> | Model | Training BS $\times$ GPU | Training Mem | Inference Mem | Video GFLOPs | Text GFLOPs |
> | :--- | :---: | :---: | :---: | :---: | :---: |
> | B/32 | 24 $\times$ 32 | 30.2 GB | 3.3 GB | 69 | 3 |
> | B/16 | 12 $\times$ 64 | 27.5 GB | 4.5 GB | 278 | 3 |
> | H/14 | 2 $\times$ 80 | 31.3 GB | 8.8 GB | 2650 | 24 |
>
> **Q2: How does the model perform under conditions of noisy or incomplete data?  Consider evaluations that specifically test the model's robustness.**
>
> **A:** Thanks for the constructive advice! To test the robustness of our model, we randomly mask some patches in each frame during inference to simulate corrupted data. The retrieval performance under different masking ratios on MSR-VTT is reported in Appendix Table  18. Thanks to the masked contrastive pre-training, our models are robust to data corruption. Impressively, even minor data distortions, such as 10% or 20% masking, do not significantly impact performance. More notably, our models demonstrate commendable robustness, maintaining considerable effectiveness even under substantial data incompleteness, with a masking ratio as high as 50%. This clearly underscores the reliability and adaptability of our models in handling corrupted data scenarios.
>
> | Model | Masking Ratio | R@1 | R@5 | R@10 | MdR |
> | :--- | :---: | :---: | :---: | :---: | :---: |
> | B/16 | 0 | 35.9 | 61.2 | 71.3 | 3.0 |
> | | 10 | 35.4 | 61.1 |71.0 | 3.0 |
> | | 20 | 35.0 | 60.6 | 70.4 | 3.0 |
> | | 50 | 32.6 | 60.0 | 69.9 | 3.0 |
> | H/14 | 0 | 38.2 | 62.4 | 73.2 | 3.0 |
> | | 10 | 37.3 | 62.1 | 72.7 | 3.0 |
> | | 20 | 36.0 | 61.6 | 72.1 | 3.0 |
> | | 50 | 35.2 | 60.4 | 70.1 | 3.0 |

---

> > ### Author Response · Authors · 2023-11-22
> > **Looking forward to your feedback**
> >
> > Thanks for your insightful comments and acknowledgment of our work. Please kindly let us know if our response addressed your concerns. We are willing to respond to any further questions before the rebuttal ends **(within 36 hours)** :-)

---

### Official Review · Reviewer_NPSQ · 2023-11-01

**Soundness:** 3 good
**Presentation:** 3 good
**Contribution:** 2 fair
**Rating:** 5
**Confidence:** 4

**Summary:**

In this paper, the authors focus on building large-scale robust video models for out-of-the-box usage, which means the learned features can be used directly for novel tasks.

The authors have conducted detailed experiments and found that tuning text encoder end-to-end causes overfitting, thus losing generalization ability. To fix the issue, they propose to tune the text encoder partially.

Finally, they adopt the transcript sorting task and masking techniques to scale up pretraining. The 1B model achieves new SOTA results on out-of-the-box tasks.

**Strengths:**

- The paper is well-written and organized, with clear figures and tables.
- The logic is clear and easy to follow.
- Extensive ablation studies and analysis demonstrate the authors' statements.

**Weaknesses:**

Overall, I appreciate the simple yet effective techniques in this paper. However, considering the differences between TVTSv2 and TVTSv1, the current paper may not be suitable for a conference but a journal as an extension:

1. Different tuning: The key difference between the two versions is how to tune the text encoder. It is an interesting finding but may not be a qualified novelty for a new conference paper. And the poor performances caused by the weak initialization (Appendix D)?
2. Same objectives/architecture/masking: The transcript sorting task and masking techniques have also been used in TVTSv1, though the masking strategies are different. And the architectures are the same (as Frozen[1]), where the residuals are skip-connected.

Considering the minor difference, I suggest the authors submit the paper as an extensive journal paper, but not a novel conference paper.

----
Reference:

[1] Bain, Max et al. “Frozen in Time: A Joint Video and Image Encoder for End-to-End Retrieval.” ICCV 2021.

**Questions:**

1. For DiDeMo, the code shows that it was tested on test split. Should it be tested on validation split?
2. In Table 2, why the single-stream models are de-emphasized?
3. In Table 2, should the authors list the size of the pretraining data for a clear comparison? It seems that the results for UMT are on a small 5M data, but others are on larger data.
4. In Table 2, why the authors do not consider ViT-L and directly scale it to ViT-H?
5. In Table 3, do those models good at retrieval also perform well, like OmniVL, CLIP-ViP, and UMT?

----
Reference:

[1] Wang, Junke et al. “OmniVL: One Foundation Model for Image-Language and Video-Language Tasks.” NeuIPS 2022.

[2] Xue, Hongwei et al. “CLIP-ViP: Adapting Pre-trained Image-Text Model to Video-Language Representation Alignment.” ICLR2023.

[3] Li, Kunchang et al. “Unmasked Teacher: Towards Training-Efficient Video Foundation Models.” ICCV2023.

---

> ### Author Response · Authors · 2023-11-19
> **Response to Reviewer NPSQ**
>
> **W1: Minor difference between TVTSv1 and TVTSv2**
>
> **W1.1: The partially frozen tuning strategy lacks novelty.**
>
> **A:** Thanks, we are willing to emphasize the novelty of the proposed partially frozen tuning. Note that it plays a vital role (rather than a trick) in training on the easy-to-scale-up ASR-video pairs, which is not trivial given that few previous papers consider tackling the text distortion issue. We carefully break down the experiment step and pinpoint the problem. We identify the problem via comprehensive experiments and figure out a simple yet effective solution that is in favor of practical use and can be well transferred to different datasets (Table 6) and vision models (Table 7). The logic, motivation and ablation studies are mostly acknowledged by all reviewers.
>
> To further underscore our contribution, we introduce a ViT-B/32-based alternate baseline, integrating adapters [1] between text encoder layers, denoted as M$_{\text{adapter}}$. During training, only the adapter-introduced parameters are trainable. The MSR-VTT retrieval results, detailed in the subsequent table, show that this alternative method yields comparatively lower performance. This outcome reinforces our assertion that end-to-end tuning can cause style overfitting, even when the text encoder remains unaltered. It also confirms that our approach of partially frozen tuning is not just a superficial fix, but a substantive and effective solution to the identified problem. We have added this discussion in our revised manuscript (Appendix D Table 16).
>
> | Method | R@1 | R@5 | R@10 | MdR |
> | :--- | :---: | :---: | :---: | :---: |
> | CLIP | 30.6 | 54.4 | 64.3 | 4.0 |
> | M$_{\text{ours}}$ | 34.5 | 58.5 | 67.7 | 3.5 |
> | M$_{\text{adapter}}$ | 28.8 | 54.8 | 65.7 | 4.0 |
>
> We believe that a paper's contribution should not be limited to proposing novel methods. Extensive empirical studies (i.e., analyzing why performance degrades), useful findings/insights (i.e., the overfitted text model), scalable training schemes (i.e., the simple yet practical partial freezing), and foundation models (i.e., TVTSv2 models that can generalize to various tasks) are also essential to the community, especially in the era of big models. Such contributions are also acknowledged by Reviewer hjUh (Task-Agnostic Focus, Novel Pre-training Strategy, etc.) and Reviewer A6Wr (Effective Pre-training Strategy).
>
> > [1] Parameter-Efficient Transfer Learning for NLP, ICML 2019
>
> **W1.2 TVTSv1's poor performances may be caused by weak initialization (Appendix D).**
>
> **A:** We admit that a strong initialization yields better performance, as reported in Appendix D Table 20. **However, naively relying on stronger initialization leads to degraded performance when pre-training on large-scale ASR-paired videos.** As shown in Table 1 in our initial manuscript (the subsequent Table) , after equipping CLIP weights, all fully tuned variants still lag behind the partially tuned competitors (L3 vs L7, L5 vs L8), and they even degrade compared to the CLIP baseline (L1 vs L3, L5), indicating the importance of the proposed partial frozen tuning strategy for producing general-purpose video representations.
>
> | No | Method | ASR | Alt-text | Kinetics-400 (ZS AR) | DiDeMo (ZS T2V) |
> | :--- | :--- | :---: | :---: | :---: | :---: |
> | L1 | CLIP | $\times$ | $\times$ | 42.7 (+0.0) | 24.7 (+0.0) |
> | L3 | M$_{\text{ASR-Full}}$ | YT-Temporal | $\times$ | 45.7 (+3.0) | 19.3 (-5.4) |
> | L5 | M$_{\text{All-Full}}$ | YT-Temporal | WebVid-2.5M | 47.9 (+5.2) | 24.2 (-0.5) |
> | L7 | M$_{\text{ASR-Partial}}$ | YT-Temporal | $\times$ | 48.1 (+5.4) | 26.8 (+2.1) |
> | L8 | M$_{\text{All-Partial}}$ | YT-Temporal | WebVid-2.5M | 50.8 (+8.1) | 29.8 (+5.1) |

---

> > ### Author Response · Authors · 2023-11-19
> > **Response to Reviewer NPSQ (cont.)**
> >
> > **W1.3: TVTSv1 and TVTSv2 share similar objectives/architecture/masking strategy.**
> >
> > **A:** Thanks. We would like to first emphasize that the central focus of our research is on scalable pre-training, not on introducing new methods or modules. Scalability is a significant and complex area of study, as acknowledged by Reviewer hjUh. There is a wide range of cutting-edge research in this field that doesn't necessarily involve new network architectures, as seen in works like VideoMAE V2 [1] and FLIP [2]. In line with this, we plan to release all our codes and pre-trained models, along with user-friendly APIs, upon acceptance. We believe that these foundational models will be a valuable asset to the community, particularly for academic labs with limited resources for large-scale experiments. The substantial contributions of our work are worthy of recognition by top-tier conferences like ICLR.
> >
> > In addition, we would like to point out that the differences between TVTSv1 and TVTSv2 are significant, for two primary reasons:
> >
> > **(i)** Our approach in TVTSv2 involves a thorough and logical analysis for selecting the best attention mechanism for video representation learning. We start with a key question: which type of attention is most effective for this purpose? While TVTSv1 uses joint attention, we introduce a new baseline in Appendix D Table 19 for comparison. This new baseline shows that simply extending the attention range can actually degrade spatial understanding. Furthermore, we explore the optimal method to initialize temporal weights. As demonstrated in Appendix D Table 19, initializing temporal weights with zeros beats its random initialization competitor, indicating that growing temporal reasoning ability out of nothing avoids hurting the well-learned spatial prior, thus more suitable for training video foundation models.
> >
> > **(ii)** We also pioneer the exploration of masked contrastive learning in large-scale pre-training in the video domain. Previous methods, like MAEs [1, 3, 4], focused on pixel-level rather than semantic-level learning, which limited their utility in learning versatile features, as evidenced by weak linear classification results in Appendix D Table 13. In contrast, TVTSv2 employs direct contrastive learning without pixel reconstruction, leading to promising results that underscore the benefits of semantic-level pre-training based on masked representations.
> >
> > > [1] VideoMAE V2: Scaling Video Masked Autoencoders with Dual Masking, CVPR 2023
> >
> > > [2] Scaling Language-Image Pre-Training via Masking, CVPR 2023
> >
> > > [3] VideoMAE: Masked Autoencoders are Data-Efficient Learners for Self-Supervised Video Pre-Training, NeurIPS 2022
> >
> > > [4] Masked Autoencoders As Spatiotemporal Learners, NeurIPS 2022
> >
> > **Q1: For DiDeMo, the code shows that it was tested on test split. Should it be tested on validation split?**
> >
> > **A:** Thanks for your concern. We follow the prior works [1, 2, 3] to conduct paragraph-to-video retrieval on the DiDeMo test split for a fair comparison. And we have double-checked the split used in their official code implementation to avoid any unfairness.
> >
> > > [1] CLIP4Clip: An Empirical Study of CLIP for End to End Video Clip Retrieval, Neurocomputing 2022
> >
> > > [2] CLIP-ViP: Adapting Image-Text Pre-training to Video-Language Representation Learning, ICLR 2023
> >
> > > [3] Bridging Video-text Retrieval with Multiple Choice Questions, CVPR 2022
> >
> > **Q2: In Table 2, why are single-stream models de-emphasized?**
> >
> > **A:** The single-stream model refers to those using a fusion module to encode multimodal entangled representations. Besides the much higher computational complexity during indexing (i.e., $O(N*M)$ for single-stream models and $O(N+M)$ for ours), the single-stream models are incapable of out-of-the-box usage, deviating from the motivation of this work. For instance, these models cannot serve for linearly probing action recognition (Appendix D Table 13) since they cannot encode video-only representations, while our pre-trained foundation models are capable of various downstream tasks without any adaptation. We have revised our manuscript for a clearer statement.

---

> > > ### Author Response · Authors · 2023-11-19
> > > **Response to Reviewer NPSQ (cont.)**
> > >
> > > **Q3: In Table 2, should the authors list the size of the pretraining data for a clear comparison? It seems that the results for UMT are on small 5M data, but others are on larger data.**
> > >
> > > **A:** Thank you for the advice. We have listed the pre-trained datasets and their scales in Appendix D Table 11. Generally speaking, models are expected to achieve better performance with an increasing number of training data given the same training recipe. However, the performance does not always improve since more data bring more noise. Moreover, video data with curated alt-text, e.g., WebVid, is generally hard to scale up. Therefore, it is important to study scalable and robust training schemes (our motivation) that can learn from large-scale noisy video data with naturally associated text knowledge, e.g., ASR transcripts.
> > >
> > > When referring to UMT, we choose the UMT-5M that was trained under a comparable data scale (6.15M) as ours (7.5M). The other variants using 17M and 25M corpus are not considered as relatively fair competitors in our work.
> > >
> > > **Q4: In Table 2, why do the authors not consider ViT-L and directly scale it to ViT-H?**
> > >
> > > **A:** Our initial focus was to assess the efficacy of our proposed pre-training framework on base-sized models. Subsequently, we sought to evaluate its effectiveness when scaled up to larger models. However, this scaling presents significant challenges in terms of time and resource consumption. For instance, training the ViT-H/14 model required a substantial investment of time and resources, taking 10 days on 80 V100 GPUs (detailed information can be found in Appendix B Table 10). Given our limited computational resources, it was impractical to train both the ViT-L/14 and ViT-H/14 models simultaneously within our submission deadline. Consequently, to more effectively demonstrate scalability, we opted to train the larger ViT-H/14 model.
> > >
> > > We have since completed training on the ViT-L/14 variant and its performances are presented in the following table. These comprehensive results are now included in the updated Table 2. In summary, the ViT-L/14 model's performance aligns with the expected scaling rule, positioning itself performance-wise between the ViT-B/16 and ViT-H/14 models.
> > >
> > > | Method | MSR-VTT R@1 | MSR-VTT R@10 | DiDeMo R@1 | DiDeMo R@10 | LSMDC R@1 | LSMDC R@10 |
> > > | :--- | :---: | :---: | :---: | :---: | :---: | :---: |
> > > | Ours-B/32 | 34.5 | 67.7 | 31.2 | 68.3 | 16.1 | 38.7 |
> > > | Ours-B/16 | 35.9 | 71.3 | 33.4 | 70.6 | 16.9 | 38.2 |
> > > | Ours-L/14 | 36.9 | 72.9 | 33.9 | 71.0 | 17.1 | 40.6 |
> > > | Ours-H/14 | 38.2 | 73.2 | 34.6 | 71.5 | 17.3 | 41.4 |
> > >
> > > **Q5: In Table 3, do those models good at retrieval also perform well, like OmniVL, CLIP-ViP, and UMT?**
> > >
> > > **A:** Nice suggestion. Since OmniVL is not publicly available yet, we face challenges in conducting a timely comparison with it. On the other hand, we test the performances of the officially released CLIP-ViP [1] and UMT [2] based on ViT-B/16. In line with what was discussed in Q2, models like UMT typically depend on multimodal entangled representations from a cross-modal encoder. To make a fair and meaningful comparison, we remove the cross-modal encoder and directly use the features outputted by the text and visual branch, namely UMT$ _{\text{dual}}$. The results are reported in the following table. Similar to Table 2, CLIP-ViP degrades due to the overfitted text encoder. Notably, UMT$ _\text{dual}$ exhibits the weakest performance, reinforcing our point that single-stream models are heavily dependent on entangled multimodal representations and are not suitable for out-of-the-box usage. In contrast, our approach, which utilizes a simpler dual-stream architecture, clearly outperforms both CLIP-ViP and UMT$ _\text{dual}$, and the improvement to the CLIP baseline is also significant, making them eligible to serve as foundation video models. We have updated Table 3 to include these results.
> > >
> > > | Method | HMDB-51 | UCF-101 | K400 | K600 | SSV2-MC |
> > > | :--- | :---: | :---: | :---: | :---: | :---: |
> > > | CLIP | 43.2 | 68.9 | 48.0 | 62.4 | 29.6 |
> > > | CLIP-ViP | 41.2 | 58.9 | 37.6 | 46.7 | 35.5 |
> > > | UMT$_{\text{dual}}$ | 32.7 | 46.4 | 34.1 | 44.0 | 23.5 |
> > > | Ours-B/16 | 50.4 | 69.8 | 54.3 | 68.1 | 42.1 |
> > > | Ours-H/14 | 52.1 | 78.0 | 59.6 | 73.2 | 48.4 |
> > >
> > > > [1] https://github.com/microsoft/XPretrain/tree/main/CLIP-ViP
> > >
> > > > [2] https://github.com/OpenGVLab/unmasked_teacher/tree/main

---

> > > > ### Author Response · Authors · 2023-11-22
> > > > **Looking forward to your feedback**
> > > >
> > > > Thanks for your constructive advice. We value your comments and have made efforts to revise the paper. Please kindly let us know if our response addressed your concerns. We are willing to respond to any further questions before the rebuttal ends **(within 36 hours)** :-)

---

> > > > > ### Comment · Reviewer_NPSQ · 2023-11-22
> > > > > **Response for the authors**
> > > > >
> > > > > Sorry for the late response. As I claimed before, **I sincerely appreciate the simple yet effective techniques** and the authors's effort in my questions.
> > > > >
> > > > > I would like to first articulate my viewpoint. **For me, the paper's key finding of `partial tuning text encoder` is surprising, just like `mask pretraining` in FLIP. However, such a finding may not be enough for a conference paper. I acknowledge scalability is vital for current foundation models, but it's not well verified in this paper. For example, how about the `ViT-g` and how about `large noisy dataset (e.g., 25M/100M)`.** I understand the authors' constraints due to limited GPU resources, which hinder thorough verification. **So it's really hard for me to decide whether the paper should be accepted and I intend to consult with other reviewers for a final decision.** Currently, considering the relation between `V1` and `V2`, a  journal extension might be a more fitting avenue for this paper.
> > > > >
> > > > > ----
> > > > >
> > > > > As for the authors' responses:
> > > > >
> > > > > **(1)**  The concept of `single-stream` in the paper remains unclear to me. Could you please elucidate the meaning of `N & M` in the context of computational complexity during indexing? Additionally, why not opt for a simpler approach by utilizing the visual encoder and incorporating an additional classification head for linear probing?
> > > > >
> > > > > **(2)**  Concerning models with a `cross-modal encoder`, i.e., UMT in the table, would performance be enhanced by an additional encoder? Based on my experience with ALBEF-style architectures, the use of a matching loss via a `cross-modal encoder` typically bolsters performance.
> > > > >
> > > > > Another minor suggestion: The table orders can be adjusted for better reading in the Appendix.

---

> > > > > > ### Author Response · Authors · 2023-11-22
> > > > > > **Additional Response to Reviewer NPSQ**
> > > > > >
> > > > > > Thank you for sharing your concerns in a timely manner. Regarding the follow-up queries, we aim to provide a more explicit explanation:
> > > > > >
> > > > > > As you know, the cross-modal encoder is typically trained with a matching loss, **based on the cross-attention between visual and textual representations.** The elaborate interaction within the cross-modal encoder generally bolsters performance. However, it was exactly the paradigm that **sacrificed efficiency** and made them **incapable of producing out-of-the-box features.** The reasons are as follows:
> > > > > >
> > > > > > **(1) Efficiency:** The cross-modal encoder intertwines visual and textual representations, requiring joint visual-textual inputs. Given $N$ queries and $M$ database items, each query needs to be paired with the $M$ database items to yield $M$ discriminative representations, leading to an overall inference complexity of $O(N\times M)$. Since the output relies on the merged visual and textual representations, such models are called "single-stream" ones. In contrast, our "dual-stream" model generates independent modality-specific representations. For the same $N$ queries and $M$ items, we encode $N$ query representations and $M$ item representations separately, allowing for swift cosine similarity calculations. This method significantly reduces inference complexity to $O(N+M)$, offering a more efficient solution than single-stream models.
> > > > > >
> > > > > > **(2) Incapable of producing out-of-the-box features:** As mentioned above, single-stream models rely on entangled multimodal representations during inference. To further address this, we directly use the visual and textual representations output by UMT's video and text encoder, namely UMT$ _{\text{dual}}$. However, the resulting video-specific representations are inadequate for immediate application evidenced by UMT$ _{\text{dual}}$'s poor performance, indicating that focusing on visual-textual interaction via matching loss does not substantially improve the quality of video representations themselves.
> > > > > >
> > > > > > **Linear Probing:** Utilizing the visual encoder and incorporating an additional classification head for linear probing is a reasonable suggestion, and it is indeed one of our evaluation metrics. As reported in Appendix D Table 13, our models outperform both self-supervised and language-guided models in these tests. This further demonstrates the effectiveness and versatility of our proposed training methodology. Since such a metric still needs to tune a classifier, instead of being directly used out-of-the-box, we make the zero-shot-based evaluation a priority in our manuscript.
> > > > > >
> > > > > > **Minor suggestion:** We appreciate your helpful suggestion and will strive to rearrange the tables in the Appendix to enhance their readability :)
> > > > > >
> > > > > > **Scalability Concerns:** We totally understand your concerns, as training such large-scale models is indeed a tough task even for industrial organizations. Here we want to emphasize a few key points.
> > > > > >
> > > > > > We have 18 ablation studies in the main paper and Appendix in total to figure out the best scalable training recipe. We follow the common practice in the community (e.g., InternVideo and VideoMAE V2) to conduct ablation studies on the base-size model and directly scale up to larger models with the best recipe. Note that successfully obtaining performance gains compared to the baseline counterparts (CLIP in our work) on large models is non-trivial and already verified the effectiveness of our strategy to a large extent, as shown in the following Table.
> > > > > >
> > > > > > | Method | MSR-VTT R@1 | MSR-VTT R@5 | MSR-VTT R@10 | MSR-VTT MdR | Kinetics-400 Top-1 Acc | Kinetics-400 Top-5 Acc | SSV2-MC Top-1 Acc | SSV2-MC Top-5 Acc |
> > > > > > | :--- | :---: | :---: | :---: | :---: | :---: | :---: | :---: | :---: |
> > > > > > | CLIP-H/14 | 36.1 | 60.4 | 70.7 | 3.0 | 52.0 | 79.1 | 40.2 | 69.2 |
> > > > > > | Ours-H/14 | 38.2 | 62.4 | 73.2 | 3.0 | 59.6 | 84.1 | 48.4 | 77.0 |
> > > > > >
> > > > > > Besides, to our best knowledge, the released ViT-H/14 model is one of the largest open-sourced video models, anticipated to significantly advance the boundaries of research in this field.
> > > > > >
> > > > > > Hope this response helps. It is delightful to have a constructive discussion, and please feel free to drop any further questions :-)

---

### Author Response · Authors · 2023-11-19
**General Response to All Reviewers**

We sincerely thank all reviewers for your reviewing efforts :)

We have uploaded a revised version of our manuscript to raise your concerns. The figures, tables, and some paragraphs are reorganized for better readability. **All major revisions are highlighted using red fonts**, please refer to the revised version for the table/figure numbers mentioned in the response unless specifically claimed.

Please feel free to reach out if you have any further questions.

---

### Author Response · Authors · 2023-11-23
**A Kind Reminder Regarding Our Response**

We thank all reviewers for your time and effort in reviewing our paper. We have responded to all comments in the rebuttal. We would like to remind you that **the rebuttal period is approaching its end (within 3 hours).** If you have any other comments or questions, please let us know.

Thank you for your attention ^_^

---

### Meta-Review · Area_Chair_JbxQ · 2023-12-08

**Metareview:**

The paper provides an analysis of the overfitting issue when finetuning pretrained VL models for building a video encoder. Based on their finding that the end-to-end text encoder tuning is a source of degradation, the paper trains a new model whose text encoder is tuned only partially. The authors claim that this model realizes task-agnostic video representation learning.

The paper received two weak rejects and two accepts. While their finding of the importance of the frozen text encoder is appreciated by all the reviewers, two of them strongly think that the technical contribution of this paper is limited. They argue that the proposed freezing technique is more like an engineering trick and is not significant enough for a standalone full conference paper concerning the minor difference compared to the existing model TVTSv1. On the other hand, the other two positive reviewers raise concerns about limited experiments to claim the proposed learns a task-agnostic representation. The authors provided some additional experimental results to resolve these concerns but it seems like they are not fully addressed.

Overall, the AC believes that the drawbacks outweigh the benefits of the paper recommending rejection.

**Justification For Why Not Higher Score:**

Despite a few positive scores, some critical issues are not fully addressed during the rebuttal.

**Justification For Why Not Lower Score:**

N/A

---

### Decision · Program_Chairs · 2024-01-16

Reject